# Phosphorylation of PFKL regulates metabolic reprogramming in macrophages following pattern recognition receptor activation

Meiyue Wang[1], Heinrich Flaswinkel[2], Abhinav Joshi[3,4], Matteo Napoli [5], Sergi Masgrau-Alsina[5], Julia M. Kamper [1], Antonia Henne [6], Alexander Heinz[6], Marleen Berouti[1], Niklas A. Schmacke [1], Karsten Hiller [6], Elisabeth Kremmer[2], Benedikt Wefers [7,8,9,10], Wolfgang Wurst [7,8,9,10], Markus Sperandio [5], Jürgen Ruland [3,4], Thomas Fröhlich [1] & Veit Hornung [1]✉

Innate immune responses are linked to key metabolic pathways, yet the proximal signaling events that connect these systems remain poorly understood. Here we show that phosphofructokinase 1, liver type (PFKL), a rate-limiting enzyme of glycolysis, is phosphorylated at Ser775 in macrophages following several innate stimuli. This phosphorylation increases the catalytic activity of PFKL, as shown by biochemical assays and glycolysis monitoring in cells expressing phosphorylation-defective PFKL variants. Using a genetic mouse model in which PFKL Ser775 phosphorylation cannot take place, we observe that upon activation, glycolysis in macrophages is lower than in the same cell population of wild-type animals. Consistent with their higher glycolytic activity, wild-type cells have higher levels of HIF1α and IL-1β than $Pfkl^{S775A/S775A}$ after LPS treatment. In an in vivo inflammation model, $Pfkl^{S775A/S775A}$ mice show reduced levels of MCP-1 and IL-1β. Our study thus identifies a molecular link between innate immune activation and early induction of glycolysis.

To effectively respond to a wide range of internal and external cues, immune cells must integrate multiple signaling cascades to shape an appropriate response. Research in recent years has shown that these signaling pathways are tightly coupled to core metabolic pathways that play important regulatory roles in both the immediate immune cell functions and the long-term fate of these cells[1,2]. Modulation of glycolysis and oxidative phosphorylation is known to be particularly important for immune cell function. Aerobic glycolysis, also known as Warburg metabolism, is characterized by rapid glucose consumption and lactate formation despite oxygen being available[3]. This metabolic

[1]Gene Center and Department of Biochemistry, Ludwig-Maximilians-Universität München, Munich, Germany. [2]Faculty of Biology, Human Biology and Bioimaging, Ludwig-Maximilians-Universität München, Planegg-Martinsried, Germany. [3]TranslaTUM, Center of Translational Cancer Research, Technische Universität München, Munich, Germany. [4]Institute of Clinical Chemistry and Pathobiochemistry, School of Medicine, Technische Universität München, Munich, Germany. [5]Faculty of Medicine Biomedical Center, Cardiovascular Physiology and Pathophysiology, Ludwig-Maximilians-Universität München, Planegg-Martinsried, Germany. [6]Institute for Biochemistry, Biotechnology and Bioinformatics, Technische Universität Braunschweig, Braunschweig, Germany. [7]Institute of Developmental Genetics, Helmholtz Zentrum München, Neuherberg, Germany. [8]TUM School of Life Sciences, Technische Universität München, Freising-Weihenstephan, Germany. [9]German Center for Neurodegenerative Diseases (DZNE) site Munich, Munich, Germany. [10]Munich Cluster for Systems Neurology (SyNergy), Munich, Germany. ✉e-mail: hornung@genzentrum.lmu.de

pathway is mostly engaged by activated pro-inflammatory M1 macrophages or effector T cells[4,5], as it provides rapid energy supply. On the other hand, the more energy-efficient pathway of oxidative phosphorylation is typically adopted by immune cells in their resting state or those with anti-inflammatory functions, such as regulatory T cells or M2 macrophages[5,6]. Of note, the choice of these pathways not only pertains to meet immediate energy demands, but also reshapes associated biosynthetic pathways, such as nucleotide, amino acid, and lipid biosynthesis[2]. Moreover, rewiring of these metabolic pathways also facilitates the production of certain metabolites that possess distinctive non-metabolic functions that can support immediate effector functions or modulate gene expression[5,7].

A well-studied model in this context are macrophages that have been stimulated by pathogen-associated molecular patterns (PAMPs), such as lipopolysaccharide (LPS). Upon LPS stimulation, macrophages shift their metabolism towards aerobic glycolysis, which can be measured by a dramatic increase in their extracellular acidification rate (ECAR), which is a correlate for lactate secretion. This metabolic shift can be detected as early as 10−20 minutes after stimulation[8], and it is later accompanied by a reduction in mitochondrial oxidative phosphorylation[9]. Mechanistically, various pathways have been proposed to effectuate this metabolic rewiring. On one hand, an abundance of evidence indicates that pro-inflammatory signaling cascades induce the expression of rate-limiting components of the glycolytic pathway. In this context, a key role is attributed to the transcription factor HIF1α (hypoxia-inducible factor 1α), which governs the expression of a number of key regulators of glycolysis. These include the transporter GLUT1 for enhanced glucose uptake, enzymes of the core glycolytic pathway itself (e.g., hexokinase-2 and phosphofructokinase 2) and factors important in the metabolism and secretion of lactate (lactate dehydrogenase and the lactate transporter MCT4)[10]. HIF1α is regulated at both the transcriptional and post-translational levels, and both pathways have been found to be regulated in PRR-activated macrophages. On one hand, NF-κB[11] and mTOR[12] activation downstream of TLR signaling have been shown to increase HIF1α mRNA expression. On the other hand, HIF1α is stabilized when prolyl hydroxylase domain (PHD) enzymes, which hydroxylate HIF1α under normoxic conditions, are inhibited. A potent PHD inhibitor is the tricarboxylic acid (TCA) cycle intermediate succinate, which accumulates during proinflammatory activation of macrophages and subsequently leads to increased HIF1α activity[13]. However, as these pathways depend on de novo gene expression, they cannot explain the rapid increase in glycolysis observed upon innate immune stimulation. As such, they are more likely to be mechanisms that are important for medium- or long-term reprogramming of cells. Additionally, HIF1α-deficient macrophages do not display any changes in metabolic flux when studied at an early to intermediate time frame[14].

Conceptually, rapid modulation of glycolysis should concern the four flux-controlling steps of glycolysis, which regulate glucose import, hexokinase activity, phosphofructokinase I (PFK1) activity, and the export of lactate[15]. It is plausible that their activity could be enhanced by allosteric regulation, e.g., by post-translational modifications, alteration of their subcellular localization, or their stabilization. Along these lines, a rapid mode of increasing glycolytic flux that has been suggested is the Akt-dependent phosphorylation of HK-II (hexokinase II). In this model, TBK1/IKKε-dependent signaling downstream of TLRs results in the activation of Akt, which in turn phosphorylates HK-II. This phosphorylation event does not enhance enzyme activity per se, yet it promotes the association of HK-II with mitochondria[16], which is believed to enhance ATP supply to this critical rate limiting enzyme of upper glycolysis. Further, an allosteric mode of enhancing glycolysis that has been identified is the activation of PFKFB3 downstream of LPS stimulation[17]. PFKFB3 is an isoenzyme of the phosphofructokinase II (PFK2) enzyme family that catalyze the formation of fructose-2,6-bisphosphate (F2,6BP), which is a potent allosteric activator of PFK1. AMPK has been shown to phosphorylate and thereby activate PFKFB3, and studies with macrophages suggest that this also occurs upon LPS stimulation[18]. While PFKFB3 is the predominantly expressed PFK2 isoenzyme in macrophages, it is only upregulated to sufficient amounts upon pro-inflammatory signaling inputs[17,18]. This precludes this pathway having an important role in the early, transcriptionally independent rewiring of macrophage metabolism. Furthermore, non-canonical STAT3 activation has been found to be necessary for LPS-induced metabolic reprogramming in peritoneal macrophages[19]. According to this particular model, LPS-induced Ser727 phosphorylated STAT3 translocates to the mitochondria where it enhances the activity of the electron transport chain[20] and is crucial for LPS-induced increases in the TCA cycle and OCR. However, it does not appear to be critical for the induction of glycolysis, as lactate levels are found to be similar in macrophages expressing either the wild-type STAT3 or the STAT3 mutant after treatment with LPS[19].

In light of the limited understanding in the current literature, we here set out to elucidate the interplay between innate immune signaling and metabolic rewiring of macrophages. Building on the hypothesis that PRR-pathways may rapidly influence the activity of rate-limiting enzymes of glycolysis through allosteric mechanisms, we concentrate our efforts on exploring potential posttranslational modifications affecting these enzymes. Our study demonstrates that innate immune stimulation rapidly induces the phosphorylation of PFKL at Ser775 in macrophages. This modification enhances PFKL activity, resulting in a swift increase in glycolysis and the promotion of pro-inflammatory cytokine production. Given PFKL's expression pattern, the activation through its phosphorylation likely extends beyond innate immune cells, suggesting a broader role in metabolic adaptation.

## Results

### Innate immune stimulation induces PFKL phosphorylation at Ser775

Previous work has documented a robust and rapid increase in glycolytic activity in macrophages upon innate immune stimulation, e.g., upon the engagement of TLRs[8] (Supplementary Fig. 1a). To obtain a detailed picture of innate immune stimulation-induced glycolysis, we performed glycolytic stress tests in murine bone marrow-derived macrophages (mBMDM) after 1 h treatment with various TLR agonists and recombinant TNF. Our investigation revealed a significant increase in ECAR following glucose injection in both unstimulated and stimulated cells. Notably, stimulated cells exhibited a significantly higher ECAR compared to unstimulated cells. Furthermore, the introduction of oligomycin, a potent inhibitor of mitochondrial ATP synthesis, further accentuated the increased ECAR in PAMP- and TNF-stimulated cells. (Fig. 1a). Similar findings were observed in human monocyte-derived macrophages (hMDM) (Supplementary Fig. 1b). This rapid increase in glycolysis suggested that TLR signaling might induce a post-translational modification in one of the rate-limiting enzymes of the glycolytic pathway, acting as an allosteric switch. To explore this hypothesis, we analyzed a previously published phosphoproteome dataset of mBMDM stimulated with LPS[21]. Here, we found that PFKL ranked among the top ten proteins in mBMDM showing increased phosphorylation (Fig. 1b). Specifically, a peptide encompassing the C-terminal tail of PFKL was found to be robustly phosphorylated at serine residue 775 in macrophages after 30 minutes of LPS treatment (Fig. 1b). PFKL is part of the PFK1 enzyme complex, which is responsible for the first committed step of the glycolytic pathway, catalyzing the conversion of fructose 6-phosphate (F6P) and ATP to fructose 1,6-bisphosphate (F1,6BP) and ADP (Fig. 1c). In mammals, PFK1 functions as a homo- or hetero-tetramer formed by three different isoforms[22], encoded on separate gene loci: PFKP (platelet), PFKL (liver), and PFKM (muscle). While these three isoforms share high sequence identity, they differ in their C-terminal extensions, which have been implicated

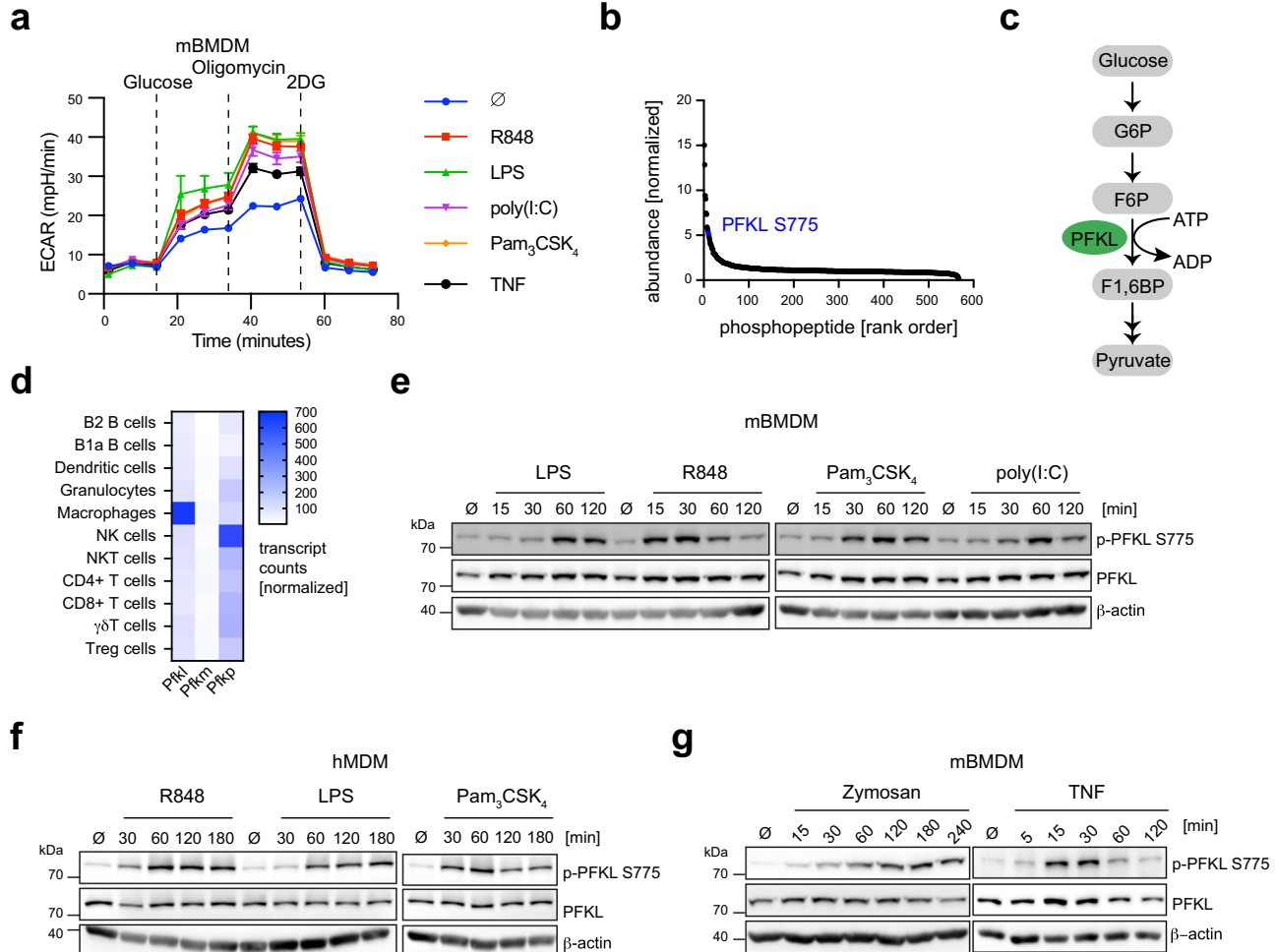

**Fig. 1 | TLR activation induces PFKL Ser775 phosphorylation in primary macrophages. a** mBMDM were stimulated with the indicated stimuli for 1 h and ECAR was then measured over time in response to the indicated compounds. Glucose (10 mM); Oligomycin (1.5 μM); 2DG (50 mM). Measurement of one representative experiment is shown. Data are presented as mean ± SEM (∅, *n* = 6; other stimuli, *n* = 8 technical replicates). **b** PFKL was identified as one of the top 10 proteins with increased phosphorylation after 30 min of LPS treatment in immortalized mBMDM[21]. **c** PFKL catalyzes the conversion of F6P to F1,6BP in the glycolytic pathway. **d** Normalized transcript counts of PFK1 isoforms in immune cells.

mBMDM (**e**) and hMDM (**f**) were stimulated with TLR agonists for the indicated time periods. PFKL Ser775 phosphorylation and PFKL were analyzed by western blot. **g** mBMDM were primed overnight with 20 ng/ml mouse IFNγ and then stimulated with zymosan (100 μg/ml) for the indicated time periods (left panel). mBMDM were stimulated with mouse TNF (50 ng/ml) for the indicated time periods (right panel). R848 (TLR7, 1 μg/ml), LPS (TLR4, 200 ng/ml), Pam₃CSK₄ (TLR1/2, 500 ng/ml), poly(I:C) (TLR3, 20 μg/ml). β-actin was used as a loading control. **a** and **f** are representative of two independent experiments. **e** and **g** are representative of three independent experiments. Source data are provided as a Source Data file.

in regulation of their activity[23]. Interestingly, the Ser775 residue within PFKL is highly conserved across vertebrates (Supplementary Fig. 1c). In addition, PFKL is the predominantly expressed isoform of PFK1 in macrophages (Fig. 1d), suggesting that PFKL may play a critical role in controlling glycolysis in the context of innate immune responses.

To study whether macrophage activation indeed triggers the phosphorylation of PFKL at Ser775, we raised antibodies against a C-terminal PFKL peptide specifically containing phosphorylated Ser775 or its unphosphorylated counterpart. Thus-obtained monoclonal antibodies generated a specific signal to either Ser775 phosphorylated PFKL or total PFKL in immunoblot. Studying the level of PFKL Ser775 phosphorylation via immunoblot revealed a rather weak signal in resting mBMDM (Fig. 1e). However, upon treatment of mBMDM with various TLR ligands such as LPS, R848, Pam₃CSK₄, and poly(I:C), there was a significant increase in PFKL Ser775 phosphorylation, while total PFKL levels remained unchanged (Fig. 1e). Analogous findings were also made in hMDM. Resting hMDM displayed little to no phospho-Ser775 PFKL signal, while stimulation with R848, LPS, or Pam₃CSK₄ led to a robust increase in PFKL Ser775 phosphorylation (Fig. 1f). Furthermore, zymosan, which activates both TLR2 and

dectin-1, and TNF also resulted in a significant increase in PFKL Ser775 phosphorylation in mBMDM (Fig. 1g). Collectively, these findings demonstrate that TNF and TLR activation induces PFKL phosphorylation at Ser775 in primary macrophages, which may be a crucial mechanism for TNF and TLR-induced early glycolytic burst.

## PFKL Ser775 phosphorylation is regulated by the IKK complex or PKCδ

One of the apical events of TLR signaling is the activation of the TAK1 and IKK kinase complexes, which are activated sequentially involving the protein kinases TAK1 and IKKβ. Furthermore, it has also been suggested that the PI3K-Akt axis plays an important role in driving the glycolytic response in dendritic cells[24], although the precise signaling events downstream of TLR activation engaging this pathway are less well understood. To investigate the involvement of these kinase complexes, we treated mBMDM with specific TAK1 (Takinib), IKKβ (TPCA-1), PI3K (Wortmannin) or AKT (MK2206 2HCl) inhibitors and then studied PFKL Ser775 phosphorylation following TLR stimulation. To test the activity and specificity of these inhibitors, we additionally probed for the phosphorylation of p65 downstream of the IKK

complex (for Takinib and TPCA-1), as well as the phosphorylation of AKT at residues T308 and S473. Doing so revealed that PFKL Ser775 phosphorylation was markedly impaired when TAK1 and IKKβ were inhibited, while the reduction in PFKL phosphorylation was paralleled by a comparable drop in p65 phosphorylation (Fig. 2a, b). AKT phosphorylation was similarly attenuated when these inhibitors were used (Fig. 2a, b). On the other hand, blocking PI3K and AKT activation had no impact on PFKL phosphorylation levels, despite the fact that AKT phosphorylation was largely blocked (Fig. 2c, d). Studying human macrophages produced similar results. As such, increasing doses of Takinib and TPCA-1 completely blunted PFKL phosphorylation, as well as AKT phosphorylation (Fig. 2e, f). Conversely, Wortmannin had no impact on PFKL phosphorylation despite fully blocking AKT phosphorylation (Fig. 2g). TAK1 and IKKβ inhibition also suppressed PFKL

phosphorylation in response to stimulation with zymosan, which activates both TLR2 and dectin-1 (Fig. 2h). In order to probe for the specific role of dectin-1 signaling, we additionally used Go 6983 to block protein kinase C (PKC), which is a critical signaling component downstream of the dectin-1 pathway[25]. Treatment of zymosan-stimulated cells with Go 6983 reduced in PFKL Ser775 phosphorylation to a similar extent as IKK inhibition (Fig. 2i), while it did not affect PFKL phosphorylation induced by other TLR agonists (Supplementary Fig. 2a).

In order to verify the participation of IKKβ and PKC in the phosphorylation of PFKL Ser775, we also examined the phosphorylation of PFKL Ser775 in a gain-of-function setting overexpressing either IKKβ or PKCδ. PKCδ was selected for this study due to its critical function in the zymosan-induced activation of primary monocytes and mBMDM[25,26].

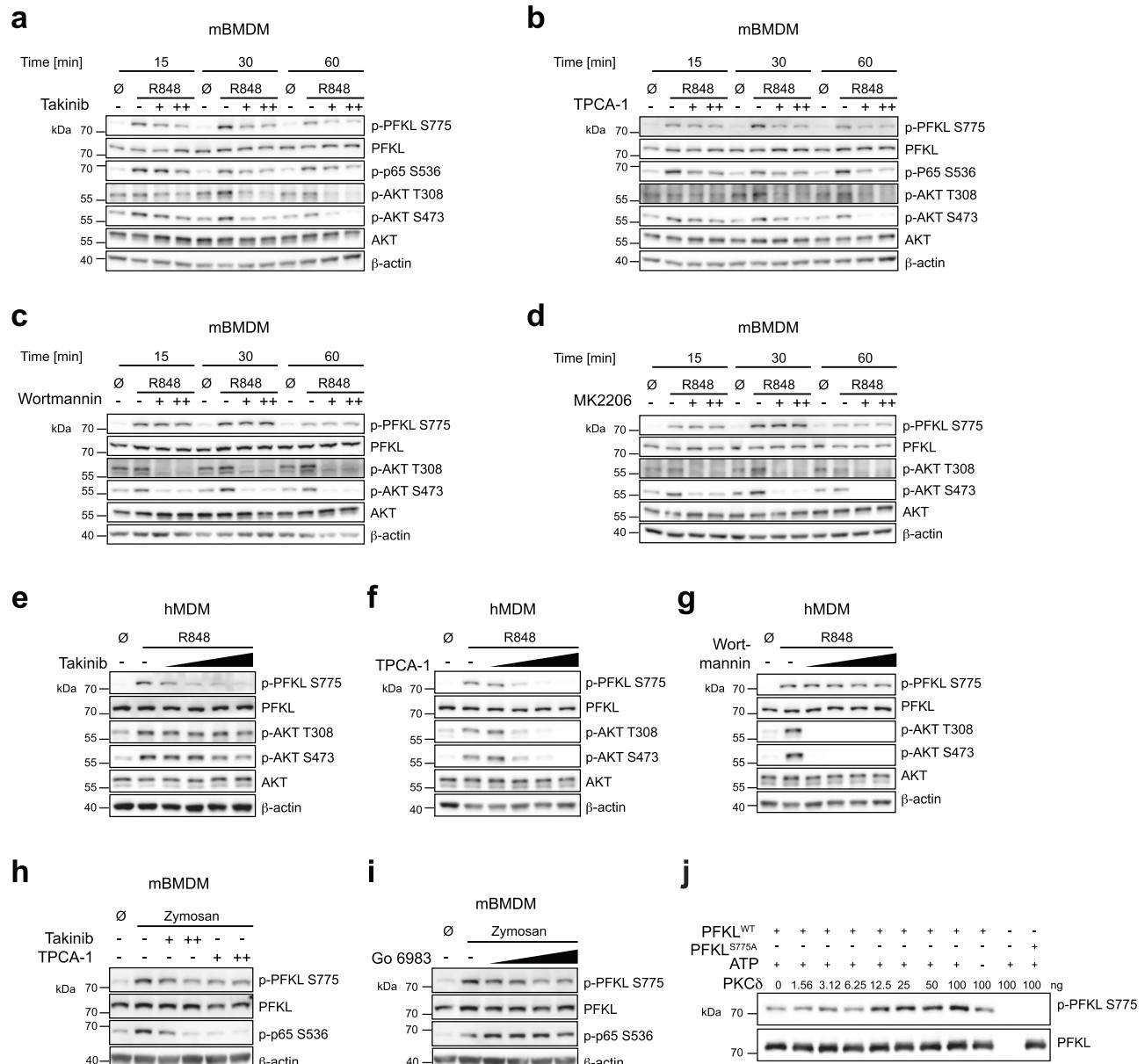

**Fig. 2 | The IKK complex and PKC function in PFKL Ser775 phosphorylation.** Immunoblot analysis of indicated proteins. mBMDM were pretreated with DMSO, Takinib (+, 50 μM; ++, 100 μM) (**a**), TPCA-1 (+, 5 μM; ++, 10 μM) (**b**), Wortmannin (+, 10 μM; ++, 20 μM) (**c**), or MK2206 2HCl (+, 10 μM; ++, 20 μM) (**d**) for 1 h and then stimulated with R848 (1 μg/ml) for the indicated time periods. hMDM were pretreated with DMSO, Takinib (1, 5, 10 and 20 μM) (**e**), TPCA-1 (1, 5, 10 and 20 μM) (**f**), or Wortmannin (1, 5, 10 and 20 μM) (**g**) for 1 h and then stimulated with R848 (1 μg/

ml) for 1 h. **h**, **i** mBMDM, primed with 20 ng/ml mouse IFNγ overnight, were pretreated with DMSO, TPCA-1 (+, 1 μM; ++, 5 μM), or Takinib (+, 1 μM; ++, 5 μM) (**h**), or Go 6983 (0.1, 1, 5 and 10 μM) (**i**) for 1 h and then stimulated with zymosan (100 μg/ml) for 1 h. **j** PFKL Ser775 phosphorylation and total PFKL levels were analyzed using western blot after the in vitro kinase assay. **a–j** are representative of three independent experiments. Source data are provided as a Source Data file.

Overexpressing these constructs in HEK293T showed that IKKβ induced PFKL Ser775 phosphorylation in a dose-dependent manner (Supplementary Fig. 2b). The induction was independent of PKC since Go 6983 did not affect this process (Supplementary Fig. 2c). While PKCδ overexpression alone caused only a slight increase in PFKL Ser775 phosphorylation (Supplementary Fig. 2b), the addition of PMA, a PKC activator, significantly boosted PKCδ-induced PFKL Ser775 phosphorylation (Supplementary Fig. 2d), suggesting that transiently expressed PKCδ is in an autoinhibitory state, requiring upstream signals to get activated[27]. Furthermore, the effect was solely reduced by Go 6983, but not TPCA-1 (Supplementary Fig. 2d), indicating that PFKL Ser775 phosphorylation triggered by PKCδ relies on a distinct pathway from IKKβ.

Using a kinase phosphorylation prediction tool suggested that both PKCδ and IKKβ were highly likely kinases for PFKL Ser775[28]. Among the 303 kinases considered, PKCδ ranked as high as 9 for the mouse PFKL Ser775 site and 15 for the human PFKL Ser775 site (Supplementary Data 1). Therefore, we further validated whether PKCδ directly phosphorylates PFKL using an in vitro kinase assay with recombinant PKCδ and human PFKL. Immunoblot analysis revealed a dose-dependent increase in the phosphorylation level of wild-type PFKL (PFKL$^{WT}$) at Ser775 upon PKCδ addition (Fig. 2j). Conversely, no signal was detected for a PFKL point mutant in which Ser775 was substituted with alanine (PFKL$^{S775A}$) at the highest dose of PKCδ (Fig. 2j). Taken together, these results suggest that TLR-induced PFKL phosphorylation occurs downstream of the IKK complex, independent of AKT signaling. Moreover, in the context of dectin-1 stimulation, PKCδ is required for PFKL phosphorylation at Ser775. Further, we could establish that PKCδ directly phosphorylates PFKL at Ser775.

### Phosphorylation of Ser775 enhances PFKL catalytic activity

Our hypothesis was that phosphorylation of Ser775, located in the regulatory domain of PFKL, affects its activity since the regulatory domain is known to modulate PFKL kinase activity[23]. To address this assumption, we wished to study the impact of a non-phosphorylatable PFKL mutant on glycolytic activity. To do so, we deleted PFKL in HEK293T cells (PFKL$^{-/-}$, Fig. 3a) and subjected these cells to a glycolytic stress test. PFKL$^{-/-}$ cells displayed a normal basal ECAR, but their glycolytic rate following glucose addition was significantly reduced compared to WT cells (Fig. 3b). Inhibition of mitochondrial ATP production using oligomycin did not further increase ECAR under these conditions, yet displayed a similar defect in glycolytic rate for the PFKL$^{-/-}$ cells (Fig. 3b). In these cells, we then re-expressed either PFKL$^{WT}$ or PFKL$^{S775A}$ using a doxycycline-inducible system. Although both PFKL$^{WT}$ and PFKL$^{S775A}$ were expressed at comparable levels (Fig. 3c), PFKL$^{-/-}$ cells expressing PFKL$^{S775A}$ showed a significantly reduced glycolytic activity compared to PFKL$^{WT}$ expressing cells (Fig. 3d, e). These results indicated that phosphorylation of Ser775 enhances PFKL activity and consequently increases glycolysis. To study PFKL$^{WT}$ and PFKL$^{S775A}$ enzymatic activity in vitro, we expressed these enzymes in HEK293T cells and then purified them using an affinity tag (Supplementary Fig. 3a). We then measured their catalytic activity by determining F1,6BP production over time using liquid chromatography-mass spectrometry (LC-MS) (Supplementary Fig. 3b). By comparing the initial reaction rates, we found that the enzymatic activity of the PFKL$^{S775A}$ mutant was approximately half of its wildtype counterpart (Fig. 3f). In order to estimate the amount of phosphorylated PFKL$^{WT}$ enzyme under these heterologous expression conditions, we conducted immunoprecipitation experiments using our phospho-specific Ser775 PFKL antibody and determined the amount of total PFKL that was not bound by the p-Ser775 PFKL antibody (Fig. 3g). Doing so, we found that approximately 30% of the PFKL$^{WT}$ enzyme was phosphorylated (Fig. 3h). In summary, our results indicate that phosphorylation of Ser775 enhances PFKL activity, providing a mechanistic basis for the regulation of PFKL activity following innate immune stimulation.

### Generation and characterization of a Pfkl S775A mouse model

To explore the role of PFKL Ser775 phosphorylation in primary cells and in vivo, we engineered a mouse model in which PFKL can no longer undergo phosphorylation at Ser775. To do so, PFKL Ser775 was mutagenized to Ala775 in murine blastocysts using CRISPR/Cas9 technology. The resulting mice were bred to contain bi-allelic mutations of Ser775Ala PFKL (Pfkl$^{S775A/S775A}$) (Fig. 4a). Compared to their WT counterparts, Pfkl$^{S775A/S775A}$ mice did not show any gross developmental defects and there were also no differences in body or spleen weight between the two groups (Fig. 4b–d). To study the quantity and composition of immune cells in both WT and Pfkl$^{S775A/S775A}$ mice, we conducted flow cytometry analysis of spleen and bone marrow cells. These studies revealed that the percentages of CD4$^+$ T cells (Supplementary Fig. 4a), NKp46$^+$ NK cells (Supplementary Fig. 4b), and CD11c$^+$MHCII$^+$ dendritic cells (Supplementary Fig. 4c) were all significantly decreased in the bone marrow of Pfkl$^{S775A/S775A}$ mice compared to that of WT mice. On the contrary, the percentages of CD11b$^+$Ly-6G$^+$ neutrophils were significantly increased (Fig. 4e, f), while the percentages of CD8$^+$ T cells (Supplementary Fig. 4d), CD19$^+$ B cells (Supplementary Fig. 4e), and CD11b$^+$F4/80$^+$ macrophages (Supplementary Fig. 4f) remained unaffected. Spleens of Pfkl$^{S775A/S775A}$ mice displayed a significantly higher proportion of NKp46$^+$ NK cells (Supplementary Fig. 4g), CD11b$^+$F4/80$^+$ macrophages (Supplementary Fig. 4h) and CD11b$^+$Ly-6G$^+$ neutrophils (Fig. 4g, h) compared to WT mice, whereas no significant differences were observed for CD4$^+$ T cells (Supplementary Fig. 4i), CD8$^+$ T cells (Supplementary Fig. 4j), CD19$^+$ B cells (Supplementary Fig. 4k), and CD11c$^+$MHCII$^+$ dendritic cells (Supplementary Fig. 4l). We also measured several pro-inflammatory cytokines in the serum including IL-1α, IL-1β, IL-6 and IL-27. However, no difference was observed between WT and Pfkl$^{S775A/S775A}$ mice for these cytokines (Supplementary Fig. 4m–p). Altogether, these observations suggested that blunting phosphorylation-induced PFKL activity does not lead to major disturbances of the immune system, at least under the steady state conditions analyzed here. Nevertheless, a moderate, yet consistent increase in neutrophil numbers in both bone marrow and spleen indicates a potential increase of granulopoiesis in Pfkl$^{S775A/S775A}$ mice.

To investigate the role of PFKL Ser775 phosphorylation in the context of inflammation in vivo, we studied its role in an intraperitoneal TNF-injection model. Serum cytokine analysis showed significantly reduced levels of monocyte chemoattractant protein-1 (MCP-1) and IL-1β in TNF-injected Pfkl$^{S775A/S775A}$ mice compared to WT mice (Fig. 4i, j). No significant differences were detected in other cytokines such as IL-6 and TNF. Evaluation of white blood cell (WBC) counts revealed a significant depression in cell number following TNF injection in WT mice compared to mock treated mice (Fig. 4k). However, there was no discernible difference in WBC counts between TNF-injected WT and Pfkl$^{S775A/S775A}$ mice (Fig. 4k). Assessing the ratios of lymphocytes, neutrophils, eosinophils and monocytes among WBC, revealed that the ratios of lymphocytes and neutrophils were similar between TNF-injected WT and Pfkl$^{S775A/S775A}$ mice (Fig. 4l, m). However, TNF-injected WT mice exhibited a significantly lower ratio of monocytes and a higher ratio of eosinophils compared to TNF-injected Pfkl$^{S775A/S775A}$ mice (Fig. 4n, o). These findings suggest that PFKL Ser775 phosphorylation plays a role in modulating the immune response during TNF-induced inflammation, particularly in regulating MCP-1 and IL-1β levels. This is accompanied by a differential depression of peripheral blood monocyte numbers, a phenomenon that may be secondary to changes in MCP-1 levels.

### Pfkl$^{S775A/S775A}$ macrophages display a blunted glycolytic response following innate immune stimulation

To explore whether PFKL phosphorylation contributes to enhanced glycolysis upon innate immune stimulation in primary macrophages, we conducted glycolytic stress tests using mBMDM isolated from

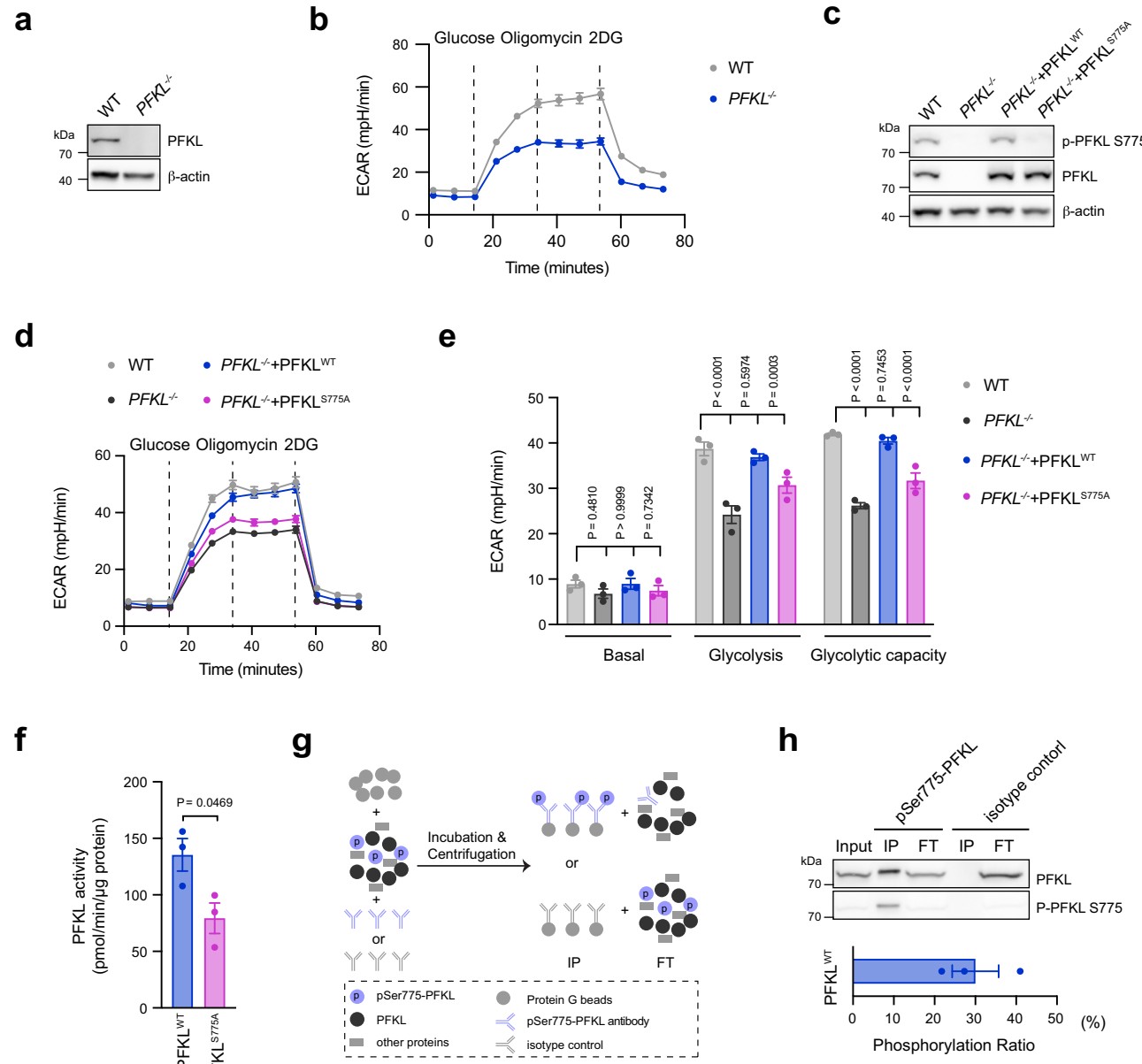

**Fig. 3 | PFKL Ser775 phosphorylation increases its catalytic activity. a** PFKL knockout in HEK293T cells was analyzed by western blot. β-actin was used as a loading control. **b** ECAR in indicated HEK293T cells was measured over time in response to indicated compounds. Glucose (10 mM); Oligomycin (1.5 μM); 2DG, 2-Deoxy-D-glucose (50 mM). Measurement of one representative experiment is shown. Data are presented as mean ± SEM (WT, $n = 5$; $PFKL^{-/-}$, $n = 6$ technical replicates). **c** PFKL knockout reconstituted with either PFKL$^{WT}$ or PFKL$^{S775A}$ was analyzed by western blot. β-actin was used as a loading control. **d** ECAR in indicated HEK293T cells was measured over time in response to indicated compounds. Measurement of one representative experiment is shown. Data are presented as mean ± SEM (WT, $n = 6$; other genotypes, $n = 8$ technical replicates). **e** ECAR in different stages of the measurement from **d** are presented as mean ± SEM ($n = 3$),

statistics indicate two-way ANOVA with Dunnett's correction for multiple testing. **f** The enzymatic activity of purified PFKL$^{WT}$ and PFKL$^{S775A}$ was measured by performing in vitro enzymatic assays and represented by the initial reaction rate. Data are presented as mean ± SEM ($n = 3$), statistics indicate unpaired two-tailed student's *t*-test. **g** Workflow for analyzing the Ser775 phosphorylation ratio of PFKL$^{WT}$ was illustrated. IP immunoprecipitated samples, FT flow through samples. **h** PFKL from the indicated fractions was analyzed by immunoblot (top panel), and the phosphorylation ratio at Ser775 of PFKL$^{WT}$ was analyzed by measuring PFKL band intensities in FT using ImageJ software (bottom panel). Data are presented as mean ± SEM ($n = 3$). Immunoblot results in **a**, **c**, **h** are representative of three independent experiments. Source data are provided as a Source Data file.

both WT and *Pfkl*$^{S775A/S775A}$ mice. No difference in glycolysis was observed between WT and *Pfkl*$^{S775A/S775A}$ macrophages in the resting state (Fig. 5a). However, after 1 h of LPS treatment, glycolysis and glycolytic capacity were reduced by about a third in *Pfkl*$^{S775A/S775A}$ macrophages relative to WT cells (Fig. 5a, b). Furthermore, following 24 h of LPS treatment, we observed that glycolysis and glycolytic capacity in *Pfkl*$^{S775A/S775A}$ cells were still decreased to approximately 80% of the levels in WT cells (Fig. 5c, d). Similar results were observed

upon stimulation with murine TNF, where WT macrophages exhibited significantly higher glycolysis and glycolytic capacity compared to *Pfkl*$^{S775A/S775A}$ macrophages following either 1 or 24 h of TNF treatment (Fig. 5e–h). To further investigate how PFKL Ser775 phosphorylation affects glycolysis in response to zymosan, we additionally conducted metabolic flux assays using a stable isotope tracer. We treated mBMDM from WT and *Pfkl*$^{S775A/S775A}$ mice with zymosan in a medium containing U-$^{13}$C-glucose and subsequently

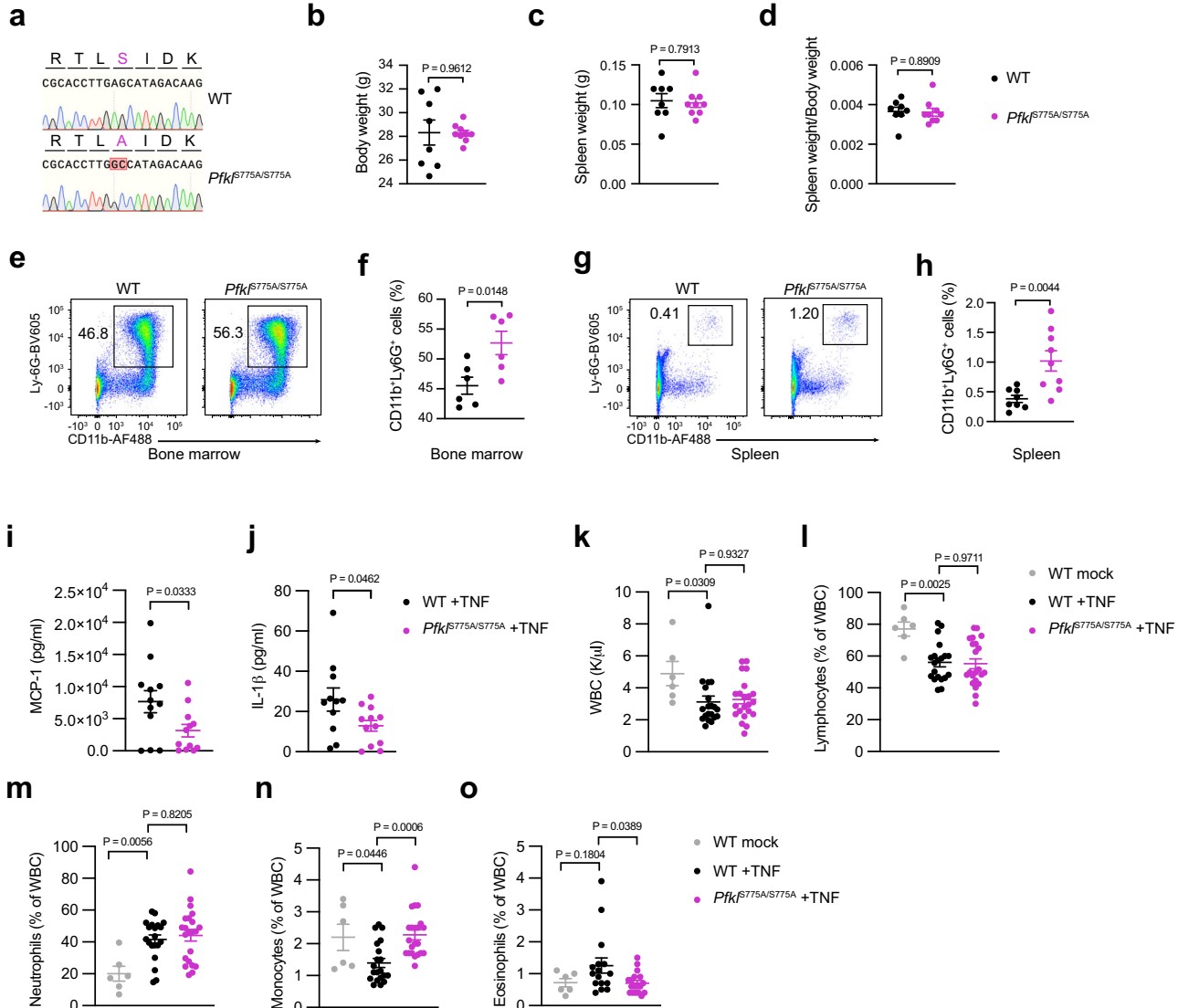

**Fig. 4 | Characterization of *Pfkl*^S775A/S775A^ mice. a** The S775A mutation in *Pfkl*^S775A/S775A^ mice was confirmed by Sanger sequencing. Body weight (**b**), spleen weight (**c**), and the ratio of spleen weight to body weight (**d**) of 10-week-old WT and *Pfkl*^S775A/S775A^ mice. Data are presented as mean ± SEM (WT mice, *n* = 8; *Pfkl*^S775A/S775A^ mice, *n* = 9), statistics indicate unpaired two-tailed student's *t*-test. **e** CD11b⁺Ly6G⁺ neutrophils from the bone marrow of WT and *Pfkl*^S775A/S775A^ mice were analyzed by flow cytometry. Numbers next to the outlined areas indicate the percentage among living cells. **f** Percentages of CD11b⁺Ly6G⁺ neutrophils from **e** are summarized and presented as mean ± SEM (*n* = 6), statistics indicate unpaired two-tailed student's *t*-test. **g** CD11b⁺Ly6G⁺ neutrophils from the spleen of WT and *Pfkl*^S775A/S775A^ mice were analyzed by flow cytometry. Numbers next to the outlined areas indicate the percentage among living cells. **h** Percentages of CD11b⁺Ly6G⁺ neutrophils from **g** are

summarized and presented as mean ± SEM (WT mice, *n* = 8; *Pfkl*^S775A/S775A^ mice, *n* = 9), statistics indicate unpaired two-tailed student's *t*-test. WT and *Pfkl*^S775A/S775A^ mice were injected with NaCl or murine TNF intraperitoneally. The levels of serum MCP-1 (**i**) and IL-1β (**j**) in TNF-injected WT and *Pfkl*^S775A/S775A^ mice. Data are presented as mean ± SEM (**i** *n* = 12; **j** *n* = 11 for WT + TNF, *n* = 12 for *Pfkl*^S775A/S775A^ + TNF), statistics indicate unpaired two-tailed student's *t*-test. WBC (**k**) and the percentage of indicated cell types among WBC (**l**–**o**) are shown. Data are presented as mean ± SEM (**k**–**n** *n* = 6, *n* = 20, *n* = 22 for WT mock, WT + TNF, *Pfkl*^S775A/S775A^ + TNF; **o** *n* = 6, *n* = 16, *n* = 18 for WT mock, WT + TNF, *Pfkl*^S775A/S775A^ + TNF), statistics indicate one-way ANOVA with Šidák's correction for multiple comparisons test. Source data are provided as a Source Data file.

quantified cellular metabolites by mass spectrometry. Following zymosan stimulation, we observed a significant increase in glycolysis in both WT and *Pfkl*^S775A/S775A^ mBMDM, as the total levels of F6P m + 6, F1,6BP m + 6, dihydroxyacetone phosphate (DHAP) m + 3, and glyceraldehyde 3-phosphate (GA3P) m + 3 that were derived from U-¹³C-glucose increased dramatically in both cell types after stimulation (Fig. 5i–l). Notably, WT cells showed higher levels of these metabolites than *Pfkl*^S775A/S775A^ cells (Fig. 5i–l), indicating a higher glycolytic rate in WT cells. Overall, these results suggest that PFKL phosphorylation at Ser775 plays an important role in augmenting glycolysis during innate immune stimulation in primary macrophages.

## *Pfkl*^S775A/S775A^ macrophages generate more ROS and display higher bactericidal activity

Recent work has shown that the activity of PFKL functions as a crucial switch in regulating the formation of ROS in macrophages[29] and neutrophils[30]. As such, boosting glycolysis through the allosteric activation of PFKL has been shown to blunt flux through the pentose phosphate pathway (PPP), thereby limiting NADPH production and thus the capacity of the phagosomal NADPH oxidase NOX2 to produce ROS[29,30]. We found zymosan, a potent inducer of ROS, was able to trigger the phosphorylation of PFKL at Ser775 (Fig. 1g), which enhances its catalytic activity (Fig. 3). To investigate the impact of PFKL Ser775 phosphorylation on ROS generation, we stimulated WT and

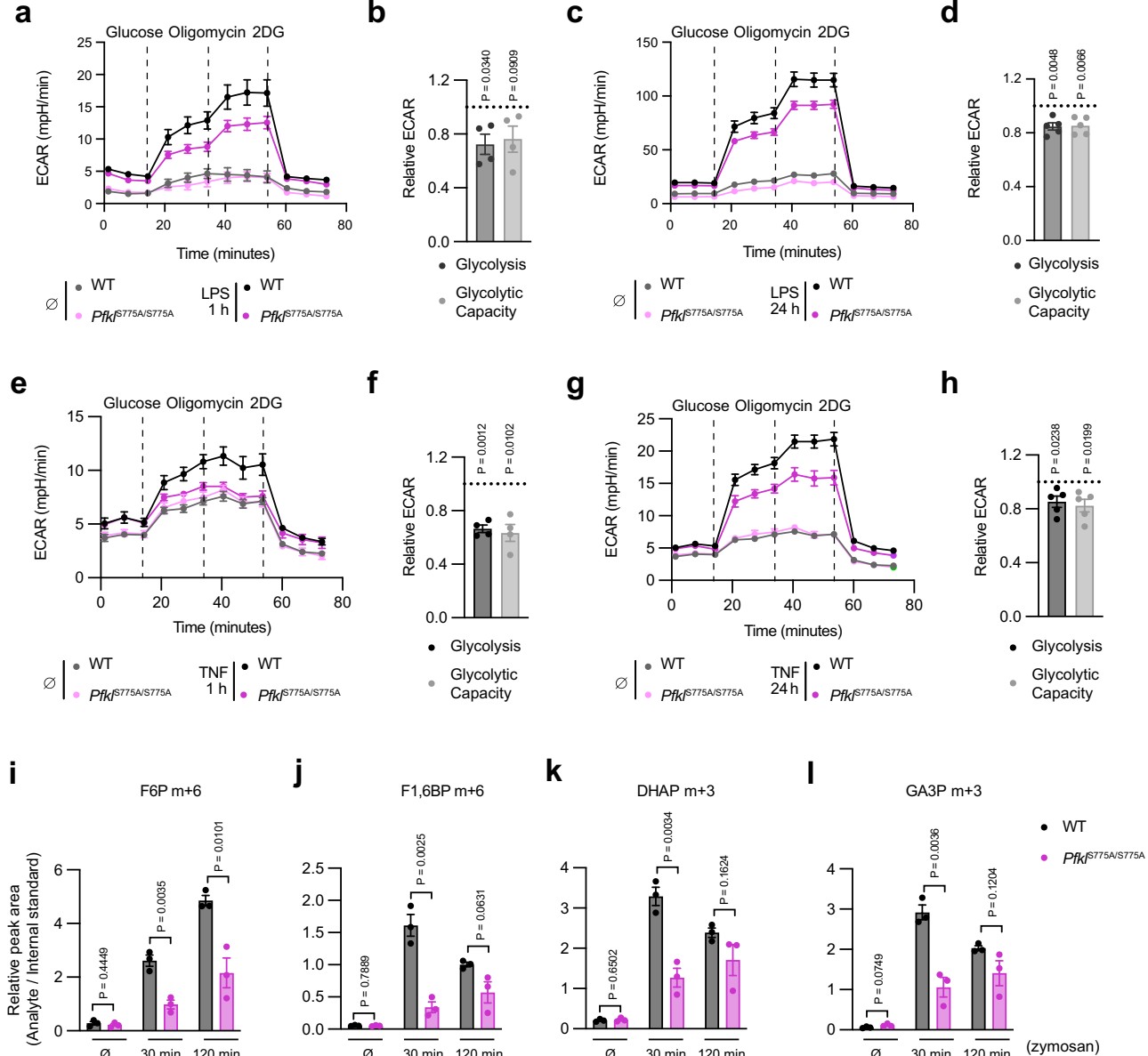

**Fig. 5 | PFKL Ser775 phosphorylation contributes to TLR-induced glycolysis in primary macrophages. a** ECAR of WT and $Pfkl^{S775A/S775A}$ mBMDM stimulated with LPS (200 ng/ml) for 1 h. Data are presented as mean ± SEM (∅, $n = 5$; LPS, $n = 6$ technical replicates). **b** ECAR of $Pfkl^{S775A/S775A}$ relative to WT at different stages of the measurement from **a** are presented as mean ± SEM ($n = 4$), statistics indicate one sample two-tailed $t$ test. **c** ECAR of WT and $Pfkl^{S775A/S775A}$ mBMDM stimulated with LPS for 24 h. Data are presented as mean ± SEM (∅, $n = 5$; LPS, $n = 6$ technical replicates). **d** ECAR of $Pfkl^{S775A/S775A}$ relative to WT at different stages of the measurement from **c** are presented as mean ± SEM ($n = 5$), statistics indicate one sample two-tailed $t$ test. **e** ECAR of WT and $Pfkl^{S775A/S775A}$ mBMDM stimulated with TNF (50 ng/ml) for 1 h. Data are presented as mean ± SEM (∅, $n = 5$; TNF, $n = 6$ technical replicates). **f** ECAR of $Pfkl^{S775A/S775A}$ relative to WT stimulated with TNF for 1 h at different stages of the measurement from **e** are presented as mean ± SEM ($n = 4$), statistics indicate one sample two-tailed $t$ test. **g** ECAR of WT and $Pfkl^{S775A/S775A}$ mBMDM stimulated with TNF for 24 h. The data for unstimulated controls in **e** and **g** were from the same experiment. Data are presented as mean ± SEM (∅, $n = 5$; TNF, $n = 6$ technical replicates). **h** ECAR of $Pfkl^{S775A/S775A}$ relative to WT stimulated with TNF for 24 h at different stages of the measurement from **g** are presented as mean ± SEM ($n = 5$), statistics indicate one sample two-tailed $t$ test. Relative abundances of F6P m + 6 (**i**), F1,6BP m + 6 (**j**), DHAP m + 3 (**k**), and GA3P m + 3 (**l**). Data are presented as mean ± SEM ($n = 3$), statistics indicate unpaired two-tailed student's $t$-test. For **a**, **c**, **e**, **g**, measurement of one representative experiment is shown. Source data are provided as a Source Data file.

$Pfkl^{S775A/S775A}$ mBMDM and measured non-mitochondrial oxygen consumption as a proxy for phagosomal ROS production. To do so, we treated cells with rotenone and antimycin A to eliminate oxygen consumption via mitochondrial respiration and then stimulated the cells with zymosan, measuring OCR in real-time. As expected, oxygen consumption strongly increased upon zymosan stimulation, peaking around 20 minutes following stimulation (Supplementary Fig. 5a). In line with previous studies showing a negative correlation between PFKL activity and ROS generation[29,30], $Pfkl^{S775A/S775A}$ macrophages

produced significantly more ROS than WT cells (Supplementary Fig. 5a, b). To address the functional relevance of this change in ROS production, we studied the bactericidal activity of WT and $Pfkl^{S775A/S775A}$ macrophages challenged with live *E. coli*. Consistent with the increased ROS production, $Pfkl^{S775A/S775A}$ macrophages demonstrated greater efficacy in killing bacteria than WT cells (Supplementary Fig. 5c). To determine whether the elevated ROS levels in $Pfkl^{S775A/S775A}$ macrophages are associated with increased NADPH production, we assessed the NADPH/NADP⁺ ratio under the specified conditions

(Supplementary Fig. 5d). Our results showed that zymosan treatment caused a reduction in this ratio in both WT and *Pfkl*[S775A/S775A] macrophages compared to resting cells (Supplementary Fig. 5e), indicating the activation of NADPH oxidase by zymosan. A slightly greater reduction was observed in *Pfkl*[S775A/S775A] macrophages compared to WT macrophages. Pretreatment with DPI, a potent NADPH oxidase inhibitor, mitigated the consumption of NADPH following zymosan treatment, resulting in an elevated ratio compared to non-pretreated cells. This ratio increased slightly more in *Pfkl*[S775A/S775A] macrophages. In summary, our data suggest that *Pfkl*[S775A/S775A] macrophages produce increased ROS levels. However, since the changes in the NADPH/NADP$^+$ ratio between WT and *Pfkl*[S775A/S775A] macrophages are only minor, we conclude that additional mechanisms beyond increased flux through the PPP contribute to increased ROS production in *Pfkl*[S775A/S775A] macrophages.

## PFKL Ser775 phosphorylation is required for LPS-induced HIF1α and IL-1β production

Glycolysis has been shown to promote pro-inflammatory functions in macrophages. One mechanism through which this occurs is glycolysis-induced HIF1α stabilization, which is specifically required for IL-1β synthesis, but not other pro-inflammatory cytokines[13]. Consistent with the notion that HIF1α is stabilized upon glycolysis, immunoblot analysis demonstrated that WT mBMDM displayed significantly higher levels of HIF1α compared to *Pfkl*[S775A/S775A] mBMDM in response to LPS treatment (Fig. 6a). Investigating the expression of *Il1b*, *Tnf* and *Il6*, we furthermore found that *Pfkl*[S775A/S775A] mBMDMs showed a significant decrease in *Il1b* expression (Fig. 6b), while *Tnf* and *Il6* expression remained intact (Supplementary Fig. 6a, c). These data were corroborated at the protein level, with pro-IL-1β levels being significantly reduced (Fig. 6c), whereas secreted amounts of TNF and IL-6 were equal between the two groups (Supplementary Fig. 6b, d). Additionally, investigating the expression of *Nos2*, another well-characterized HIF1α-target gene[31], we observed a significant reduction in its mRNA levels in *Pfkl*[S775A/S775A] compared to WT mBMDM after 24 h of LPS stimulation (Fig. 6d). It has been demonstrated that glycolysis stabilizes HIF1α by inhibiting its PHD-dependent hydroxylation which results in its degradation[32]. To confirm that the observed differences in HIF1α levels between LPS-stimulated WT and *Pfkl*[S775A/S775A] mBMDM were due to differential PHD activity, cells were pretreated with octyl-α-ketoglutarate (octyl-α-KG), which can be converted into α-KG in cells, overriding the inhibitory effects on PHDs by endogenous metabolites, such as succinate[33]. As expected, activated WT cells required more octyl-α-KG to reduce levels of HIF1α and pro-IL-1β to those observed in *Pfkl*[S775A/S775A] cells (Fig. 6e), suggesting higher PHD inhibitory activity by glycolytic metabolism in WT cells. To explore the reverse scenario, we applied DMOG, a PHD inhibitor to stabilize HIF1α. Pretreatment with DMOG caused the top band of HIF1α to increase while the bottom band vanished, indicating that the lower bands are indeed degradation products of HIF1α (Fig. 6e). WT cells generated more HIF1α in response to LPS when the same amount of DMOG was applied (Fig. 6e), in line with the notion that glycolytic metabolism and DMOG work together to stabilize HIF1α. In conclusion, these findings demonstrate that PFKL Ser775 phosphorylation plays a critical role in the stabilization of HIF1α and thus the regulation of pro-inflammatory gene expression, specifically *Il1b* and *Nos2*. Further, this activity might be mediated through glycolysis-dependent stabilization of HIF1α.

## Discussion

Although it is well known that innate immune stimulation of macrophages induces a rapid increase in glycolysis, the underlying mechanisms and its functional consequences are still unclear. Hypothesizing that a phosphorylation event might be responsible for this switch-like response, we reviewed the phosphoproteome data of TLR-stimulated macrophages and identified the PFK1 isoenzyme PFKL as a

potential candidate for stimulation-dependent modulation[21]. PFKL, which is predominantly expressed in macrophages, showed stimulation-dependent phosphorylation at Ser775, an evolutionarily well conserved site on its C-terminal tail. PFKL phosphorylation at Ser775 was dependent on the activity of the IKK complex, but independent of PI3K and AKT. Further, in the context of dectin-1 stimulation, PFKL phosphorylation required PKCδ, which we could also identify as a direct kinase for PFKL Ser775. Enzymatic activity assays revealed that Ser775 phosphorylation enhanced the catalytic activity of PFKL, while a phosphorylation-defective mutant displayed a blunted glycolytic switch response when heterologously expressed in PFKL-deficient cells. Encouraged by these findings, we generated *Pfkl*[S775A/S775A] mice in which PFKL cannot be phosphorylated at Ser775. *Pfkl*[S775A/S775A] macrophages exhibited reduced glycolysis in response to innate immune stimulation in both Seahorse and isotope tracing assays. Comparison of WT and *Pfkl*[S775A/S775A] macrophages revealed that the increase in early glycolysis was important for the expression of pro-inflammatory genes, including *Il1b* and *Nos2*. In vivo TNF-induced inflammation assays showed that PFKL Ser775 phosphorylation played a role in the positive regulation of MCP-1 and IL-1β levels. Interestingly, in addition to regulating the inflammatory response to innate immune stimuli, PFKL phosphorylation also seemed to moderately affect the composition of the immune system at steady state, as increased neutrophils were observed in both bone marrow and spleen of *Pfkl*[S775A/S775A] mice. Collectively, these results reveal a role for PFKL phosphorylation in the regulation of glycolytic flux in innate immune cells. Importantly, by relying on a targeted perturbation instead of small molecule inhibitors that are prone to off-target effects, this work provides the opportunity to explore the extent of innate-dependent enhancement of glycolysis.

PFK1 constitutes the first committed and rate-determining step of glycolysis, responsible for the phosphorylation of F6P to F1,6BP. All three PFK1 isoenzymes share considerable sequence identity in their core catalytic domains, yet differences exist with regards to regulatory domains that allow modulation of their catalytic activities based on metabolic demands[23]. The most important regulatory adaption is achieved by allosteric means, through a number of small molecule metabolites that mirror the metabolic state of the cell. Important allosteric regulators that increase enzyme activity include AMP, ADP, and F2,6BP, the latter being synthesized by PFK2 isoenzymes[23]. Less is known about posttranslational modifications regulating their activity, although GlcNAcylation has been shown to decrease enzyme activity[34]. PFK1 enzymes function as a tetramer that can be composed of different isoenzymes[22]. A distinctive property of PFKL is that it can form filamentous structures, with its C-terminal regulatory domain being essential for filament formation[35]. Our in vitro biochemical assays would suggest that PFKL Ser775 phosphorylation positively regulates PFKL catalytic activity, but the exact mechanism by which this phosphorylation regulates its activity is unknown. It is conceivable that phosphorylation affects the conformational equilibria of the tetramer between the active R-state and the inactive T-state. At the same time, it is also possible that phosphorylation influences filament formation, thereby impacting on catalytic activity. In support of our model, a systems biology approach has previously identified PFKL as a key regulatory step in insulin-dependent enhancement of glycolytic flux[36]. Developing a reconstruction method for signal flow, based on time-course phosphoproteome and metabolome data, it was found that increased PFK1 activity coincided with the Ser775 phosphorylation of PFKL. Indeed, it was then also demonstrated that phospho-mimetic PFKL mutants displayed enhanced kinase activity in comparison to the wild-type PFKL[36]. Collectively, this implies that the augmentation of PFKL activity through phosphorylation is not limited to innate immune cells, yet has a broader role in adapting metabolism.

We found that blocking IKKβ largely blunted TLR-induced PFKL Ser775 phosphorylation in primary macrophages. Conversely, AKT and

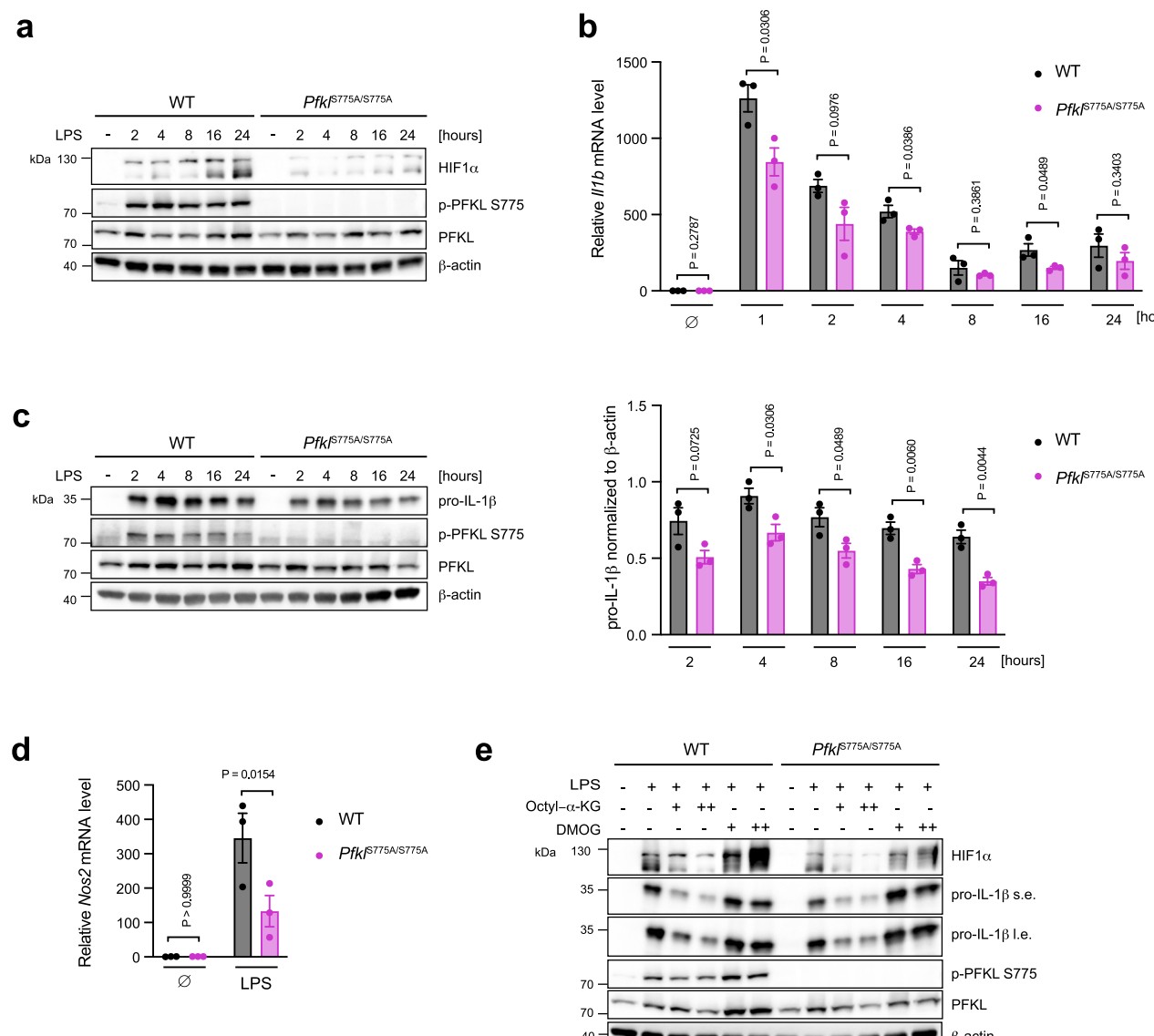

**Fig. 6 | PFKL Ser775 phosphorylation is required for LPS-induced HIF1α and IL-1β production. a** mBMDM from WT and *Pfkl*^S775A/S775A^ mice were stimulated with LPS (200 ng/ml) for the indicated time periods. Indicated proteins were analyzed by western blot. β-actin was used as a loading control. **b** LPS-induced *Il1b* mRNA levels in WT and *Pfkl*^S775A/S775A^ mBMDM were measured by qPCR and normalized to *Actb* mRNA levels. Data are presented as mean ± SEM (*n* = 3), statistics indicate unpaired two-tailed student's *t*-test. **c** mBMDM from WT and *Pfkl*^S775A/S775A^ mice were stimulated with LPS for the indicated time periods. Indicated proteins were analyzed by western blot (left panel). LPS-induced pro-IL-1β were quantified by normalization to β-actin at indicated time points (right panel). Data are presented as mean ± SEM (*n* = 3), statistics indicate unpaired two-tailed student's *t*-test. **d** mBMDM from WT

and *Pfkl*^S775A/S775A^ mice were stimulated with LPS for 24 h and *Nos2* mRNA levels were measured by qPCR and normalized to *Actb* mRNA levels. Data are presented as mean ± SEM (*n* = 3), statistics indicate two-way ANOVA with Šidák's correction for multiple comparisons test. **e** mBMDM from WT and *Pfkl*^S775A/S775A^ mice were pre-treated with octyl-α-KG (+, 0.5 mM; ++, 1 mM) or DMOG (+, 0.25 mM; ++, 0.5 mM) for 2 h and then stimulated with LPS for 24 h. Indicated proteins were analyzed by western blot. β-actin was used as a loading control. Octyl-α-KG octyl-α-ketoglutarate, DMOG dimethyloxalylglycine, s.e. short exposure, l.e. long exposure. **a**, **e** are representative of three independent experiments. Source data are provided as a Source Data file.

its upstream kinase PI3K were not involved in PFKL phosphorylation, despite the fact that TLR ligands rapidly activated AKT in an IKK-PI3K-dependent manner. These results are in contrast with previous work, in which AKT was found to be the key factor in inducing the glycolytic switch response in dendritic cells. In this model TBK1 and IKKε operated redundantly in activating AKT without the involvement of PI3K activity, which in turn resulted in HKII phosphorylation and enhancement of glycolytic flux[16]. Even though these two studies contradict each other regarding the activation of AKT, they are not mutually exclusive with regard to the activation of glycolysis. As such, it is conceivable that AKT-dependent HKII phosphorylation additionally

contributes to the glycolytic switch, in conjunction with the PFKL-dependent pathway described here.

Glycolysis has been reported to be critical for the regulation of immune functions in response to stimulation[1]. We found that *Pfkl*^S775A/S775A^ macrophages displayed lower glycolysis, which was associated with reduced IL-1β production in response to LPS treatment, whereas IL-6 and TNF production remained unchanged. This is consistent with previous work showing that LPS induced glycolysis mainly affects IL-1β production, but not generally other pro-inflammatory cytokines[13]. This has been attributed to the distinctive role of HIF1α in IL-1β expression. Mechanistically, it has been shown that the increase

in glycolysis results in PHD inhibition, which results in HIF1α stabilization and concomitant enhanced transcription of HIF1α target genes[13]. One potential mechanism is that glycolysis-associated buildup of succinate blocks PHD through a product inhibition mechanism[32]. In line with differential glycolytic activity impacting on PHD activity, more octyl-α-KG was required in activated wild-type (WT) cells to induce HIF1α destabilization. Another study has reported that an early LPS-induced increase in glycolytic flux is required for late pro-inflammatory gene expression[37]. Here it was found that TLR signaling alters flux through glycolysis and the TCA, leading to increased production of acetyl coenzyme A (acetyl-CoA), which in turn enhanced histone acetylation of certain regulatory regions of LPS-inducible gene sets[37]. A prominent target gene that was affected by this mechanism was IL-6[37]. Our observations did not show any differences in IL-6 levels between WT and $Pfkl^{S775A/S775A}$ macrophages after LPS treatment, suggesting that PFKL activity changes – as elicited by preventing its C-terminal phosphorylation–may not affect citrate and acetyl-CoA production. It is conceivable that glycolysis-dependent modulation of PHD activity is more sensitive to changes in glycolytic flux than citrate production and associated downstream events. However, more work will be required to elucidate the underlying mechanisms.

Numerous studies have provided compelling evidence that innate immune stimulation of macrophages has a profound impact on core metabolic pathways. A prominent example is the here-studied glycolytic switch that is observed in many activated immune cells, including dendritic cells, T cells, NK cells and macrophages[2,4,38]. This shift into aerobic glycolysis is well documented by readouts such as ECAR, production of specific metabolites (e.g., lactate) or even whole metabolomes. However, the key question is how functional consequences can be attributed to these changes in metabolism. A commonly used inhibitory tool compound is 2-deoxy glucose (2DG), which acts by inhibiting hexokinase, the first enzyme in the glycolytic pathway. In line with this inhibitory function, treatment of macrophages with 2DG leads to a strong decrease in their stimulation-induced glycolytic flux and lactate production[8,39]. Due to this, 2DG treatment leads to a profound destabilization of HIF1α in LPS-treated macrophages, thereby affecting the expression of HIF1α target genes, most prominently IL-1β[13]. Reassuringly, 2DG treatment has no impact on the activation of core pro-inflammatory signaling components, e.g., NF-κB, which is why it is usually considered as a specific inhibitor of glycolysis[13,40]. However, 2DG has profound off-target effects beyond inhibiting glycolysis. Among these, the most relevant appears to be the lowering of intracellular ATP levels, which compromises multiple enzymatic reactions[41,42]. In fact, culturing cells in glucose-free medium or replacing glucose with galactose, which similarly blunts the glycolytic switch response, only has a moderate impact on pro-inflammatory readouts, such as IL-1β production[43]. These results warrant a more careful interpretation of 2DG-dependent results and call for more specific approaches to study the role of metabolism in innate immune responses. In this study, our objective was to circumvent the possible off-target consequences linked to small molecule inhibitors by creating a genetic model that allows for the examination of a specific metabolic enzyme's role in macrophage function without disrupting it. This approach enabled us to reduce confounding elements and gain a clearer comprehension of the targeted pathways' genuine influence on macrophage function and immune responses, ultimately providing more reliable insights into the underlying biology. Needless to say, addressing the role of a single posttranslational modification in a complex metabolic pathway bears the risk of redundancy, as multiple modulatory steps might impact on glycolysis. Considering these factors, we posit that the moderate but significant quantitative differences we observed as a function of PFKL phosphorylation are quite plausible. At the same time, our experiments also show that additional mechanisms must exist that drive the glycolytic switch upon innate immune stimulation. To this end, while, for example, galactose-cultured macrophages show a completely blunted glycolytic switch response upon TLR stimulation, $Pfkl^{S775A/S775A}$ macrophages still showed an increase in glycolysis that was about two-thirds of that of WT cells. It is therefore conceivable that innate signals control additional steps important for enhanced glycolysis, including enhanced glucose uptake through increased externalization of Glut channels[44], increased HKII-recruitment to mitochondria[16], or other allosteric modulations on other enzymes.

## Methods

### Generation of $Pfkl^{S775A/S775A}$ mice

$Pfkl^{S775A/S775A}$ mice were generated using CRISPR/Cas9-mediated gene editing in zygotes, as previously outlined[45]. In short, pronuclear stage zygotes were obtained by mating C57BL/6 J male mice with superovulated C57BL/6 J female mice. Embryos were then electroporated using the NEPA21 electroporator and a 1 mm electrode with a $Pfkl$-specific CRISPR/Cas9 ribonucleoprotein (RNP) solution consisting of 200 ng/µl S.p. Cas9 HiFi protein (IDT), 6 µM crRNA (protospacer GAA ACCCTTGTCTATGCTCA; IDT), 6 µM tracrRNA (IDT), and 300 ng/µl mutagenic single-stranded oligodeoxynucleotide (ssODN) (5′- GGTGA GAGTCCGGGAACGCAGGAACAGTAGTGACAGTAAGCTCAGAAACCCT TGTCTATGgcCAAGGTGCGGCGTGTGACGTGCTCCAGCTCCCCAGAC ACATAGTCTGCCATGCTGATGCG-3′), comprising the S775A substitution (AGC > GCC). Post-electroporation, zygotes were implanted into pseudopregnant CD-1 surrogate animals. Potential off-target sites for the $Pfkl$-specific crRNA were predicted using the CRISPOR online tool[46] to eliminate any unwanted modifications. Genomic DNA from the F1 generation was PCR-amplified and confirmed through Sanger sequencing, revealing no additional sequence variations. F1 heterozygous mice were backcrossed with C57BL/6 J mice to produce heterozygous offspring, which were subsequently intercrossed to generate homozygous $Pfkl^{S775A/S775A}$ mice, which were then compared to WT littermate controls.

All animal experiments were approved by the Bavarian government under license number ROB-55.2-2532.Vet_02-16-121 or ROB-55.2-2532.Vet_02-20-15 (TNF injection experiments, see below). All mice were treated in compliance with the institutional guidelines approved by the animal welfare and use committee of the government of Upper Bavaria. WT and $Pfkl^{S775A/S775A}$ mice were bred separately in the same facility in standard cages in a specific pathogen-free facility, maintained on a 12-hour light/dark cycle, and given unrestricted access to food and water.

### Mouse bone marrow-derived macrophage (mBMDM) culture

Bone marrow cells were isolated from 10-12-week-old, male mice (Figs. 1, 4, Supplementary Figs. 1, 2, 4 and 5), 10–12-week-old, male and female mice (Fig. 2), and 23–25-week-old, male mice (Figs. 5, 6 and Supplementary Fig. 6). They were differentiated in Dulbecco's modified Eagle's medium (DMEM) supplemented with 10% FCS, 1% Penicillin-Streptomycin (10,000 U/ml), 1% sodium pyruvate (all from Thermo Fisher Scientific), and 30% L929 conditional media in low-adherence dishes for 6 days. On the 6th day of differentiation, mBMDM were detached with PBS containing 2 mM EDTA (Invitrogen) and replated into 12-well plates for western blot ($10^6$ cells per well). To check AKT and PFKL Ser775 phosphorylation by western blot, detached mBMDM were seeded overnight in DMEM medium supplemented with 0.5% FCS. For other experiments, mBMDM were plated in normal DMEM medium containing 10% FCS.

### Human monocyte-derived macrophages (hMDM) culture

Peripheral blood mononuclear cells (PBMCs) were isolated from the leukocyte reduction system chambers left over from platelet donation from healthy donors. Approval from the relevant ethics committee and informed consent from all donors according to the Declaration of Helsinki were obtained (project number: 19-238, Ethics Committee of

the Medical Faculty of Ludwig-Maximilians-University Munich). Human monocytes were purified from peripheral blood mononuclear cells (PBMCs) using CD14 microbeads (Miltenyi Biotec) and differentiated into monocyte-derived macrophages in RPMI 1640 (Thermo Fisher Scientific) supplemented with 10% FCS, 1% Penicillin-Streptomycin (10,000 U/ml), 1% sodium pyruvate and 200 ng/ml M-CSF (MPI of Biochemistry, Munich) for 6 days. Fresh M-CSF was added on day 3 and day 5. To check AKT and PFKL Ser775 phosphorylation by western blot, the medium was changed to RPMI 1640 medium supplemented with 0.5% FCS on day 5. Human monocytes were differentiated in Seahorse 96-well cell culture plates (Agilent Technologies) for Seahorse assay ($7 \times 10^4$ cells per well), and 12-well plates for western blot ($10^6$ cells per well).

### Generation of monoclonal antibodies against PFKL

A peptide containing amino acids HVTRRTLpSMDKGF (human PFKL) was synthesized and coupled to OVA (Peps4LS, Heidelberg, Germany). C57BL/6 mice were immunized subcutaneously and intraperitoneally with a mixture of 50 µg peptide-OVA, 5 nmol CpG oligonucleotide (Tib Molbiol, Berlin, Germany), 100 µl PBS, and 150 µl incomplete Freund's adjuvant. A re-injection without adjuvant was performed 6 weeks after the primary injection. Fusion was performed according to standard procedures. Supernatants were tested in a differential ELISA with the phosphorylated peptide or the non-phosphorylated peptide, both coupled to biotin and bound via avidin. Supernatants that reacted either independently or specifically with the phospho-peptide were further analyzed by immunoblotting to recognize human and mouse PFKL.

Corresponding hybridoma cells of selected candidates from the secondary analysis were thawed and cloned repeatedly until subclones showed complete stability. Stable clones were secured and subsequently grown for monoclonal antibody production using supernatant.

### Transfection in HEK293T cells

HEK293T cells were plated in 12-well plates at a density of $6 \times 10^5$ cells per well. 5 h later, cells were transfected with PKCδ or IKKβ using the Genejuice transfection reagent (Merck Millipore) according to the manufacturer's instructions.

### Immunoblotting

Cells were washed once with cold PBS and lysed directly on plates in DISC buffer (30 mM Tris-HCl pH 7.5, 150 mM NaCl, 1% TritonX-100, and 10% glycerol) supplemented with phosphatase inhibitor (PhosSTOP, Sigma-Aldrich) and protease inhibitor (cOmplete™ Protease Inhibitor Cocktail, Sigma-Aldrich) on ice for 10 min. Protein concentration was quantified by BCA assay (Pierce BCA Protein Assay Kit, Thermo Fisher Scientific) according to the manual instructions. Samples were mixed with 6 x Laemmli buffer and heated at 95 °C for 5–10 min and loaded on SDS-PAGE gel (Novex™ 10% Tris-Glycine Mini Gels, Thermo Fisher Scientific). Afterwards, protein was transferred onto nitrocellulose membranes (Amersham, GE Healthcare), blocked with 5% BSA, and incubated with primary antibodies overnight at 4 °C. The primary antibodies used for immunoblot included pSer775-PFKL (1:100; in house), PFKL (1:250; in house), beta-actin-HRP (1:5000; Santa Cruz sc-47778), p-NF-KB p65 (Ser536) (93H1; 1:1000; Cell signaling 3033), HA Tag mouse mAb HRP conjugate (6E2; 1:000, Cell signaling 2999 S), IKKβ (D30C6; 1:1000; Cell signaling 8943 S), p-AKT T308 (C31E5E; 1:500; Cell signaling 13038 S), p-AKT S473 (D9E; 1:1000; Cell signaling 4060 S), AKT (pan) (C67E7; 1:1000; Cell signaling 4691 S), HIF-1α (D1S7W; 1:1000; Cell signaling 36169 S), and IL-1β (1:1000; R&D system AF-401-NA).

Next day, membranes were incubated for 1 h with the appropriate secondary antibodies. The secondary antibodies used included HRP-goat anti-mouse IgG, Fcgamma secondary (1:3000; Jackson

ImmunoResearch 115-035-071), HRP -donkey anti-goat IgG (H + L) (1:3000; Invitrogen A15999) or HRP-goat anti-rabbit IgG (1:3000; Cell signaling 7074). Chemiluminescence signal was detected using a CCD camera (Fusion FX).

### Generation of knockout cells

Gene-deficient HEK293T cells were generated using CRISPR/Cas9 technology. A gRNA targeting the early coding exon of PFKL (CGCGATGTTCAGGTGCGAGTAGG) was designed and cloned into pLKO.1-gRNA-CMV-GFP plasmid. HEK293T cells were seeded in 12-well plates ($6 \times 10^5$ cells per well) for 5 h and then transfected with a plasmid expressing mCherry-Cas9 (pRZ-CMV-mCherry-Cas9) and the plasmid carrying gRNA (pLKO.1-gRNA-CMV-GFP) using GeneJuice (Merck Millipore). 24 h after transfection, mCherry positive cells were sorted and subjected to limiting dilution cloning. After 2–3 weeks, monoclones were identified, replated, and grown for genotyping by deep sequencing (Illumina's Miseq-platform). Clones carrying all-allelic frameshift mutations were used for experiments.

### Retroviral transduction

Strep-PFKL$^{WT}$ and strep-PFKL$^{S775A}$ were amplified from cDNA derived from HEK293T cells and cloned into a doxycycline-inducible plasmid (pLI_Strep_PFKL$^{WT}$_Puro and pLI_Strep_PFKL$^{S775A}$_Puro). For virus production, HEK293T cells were plated overnight in 10 cm dishes ($3 \times 10^6$ cells per dish) and transfected with viral plasmids including 12 µg pMDLg/pRRE, 4 µg pRSV-Rev, 8 µg pCMV-VSV-G, and 8 µg pLI_Strep_PFKL$^{WT}$_Puro or pLI_Strep_PFKL$^{S775A}$_Puro using calcium phosphate. The supernatant was harvested 72 h post transfection and centrifuged to remove cell debris. To introduce strep-PFKL$^{WT}$ or strep-PFKL$^{S775A}$ into PFKL$^{-/-}$ HEK293T cells, cells were infected with the corresponding virus and selected with 2.5 µg/ml puromycin for 2 days 48 h after transduction. To check the inducible expression of PFKL$^{WT}$ and PFKL$^{-/-}$ in PFKL$^{-/-}$ HEK293T cells, cells were seeded in 12-well plates ($6 \times 10^5$ cells per well) in DMEM medium and transduced with 1 µg/ml doxycycline (Sigma-Aldrich) for 24 h. The polyclonal cell population was used for experiments.

### Glycolytic stress test

HEK293T cells were seeded at a density of $2.5 \times 10^4$ per well in Seahorse 96-well cell culture plates coated with poly-L-ornithine (Sigma-Aldrich) and treated with 1 µg/ml doxycycline for 24 h. $7 \times 10^4$ mBMDM cells were plated per well and stimulated the next day with different stimuli for the indicated time periods. Cells were then washed three times with the Seahorse assay medium (Seahorse XF DMEM medium supplemented with 2 mM glutamine, all from Agilent Technologies) and cultured in this medium at 37 °C for 45 min in a $CO_2$-free incubator. Real-time respiratory and glycolytic rates were then measured in response to sequential injections by Seahorse XFe96 Analyzer (Agilent Technologies).

### PFKL purification

Transduced HEK293T cells were plated overnight in 15 cm dishes ($1 \times 10^7$ cells per dish) and then treated with 2 µg/ml doxycycline for 24 h for protein induction. Cells were washed once with cold PBS and lysed with DISC buffer supplemented with phosphatase inhibitor (PhosSTOP, Sigma-Aldrich) and protease inhibitor (cOmplete™ Protease Inhibitor Cocktail, Sigma-Aldrich) for 10 min on ice. The supernatant after centrifugation was incubated with equilibrated Strep-Tactin XT beads (IBA Lifesciences) overnight at 4 °C and proteins were eluted by incubating the beads with elution buffer containing 50 mM Tris-HCl (pH 7.8), 50 mM biotin (Sigma-Aldrich), 100 mM KCl, and 1 mM DTT (Sigma-Aldrich). The eluate was further concentrated using a 10 kDa MWCO Amicon Ultra Centrifugal Filter (Merck Millipore). The concentration of purified proteins was determined by BCA Assay and purity was determined by Coomassie staining.

## Immunoprecipitation

HEK293T cells expressing inducible PFKL$^{WT}$ were plated overnight in 15 cm dishes ($1 \times 10^7$ cells per dish) and then treated with 2 µg/ml doxycycline for 24 h. Cells were washed once with cold PBS and lysed in DISC buffer supplemented with phosphatase inhibitor (PhosSTOP, Sigma-Aldrich) and protease inhibitor (cOmplete™ Protease Inhibitor Cocktail, Sigma-Aldrich) for 10 min on ice. Cell lysate was collected after centrifugation and incubated with 100 µl of the pSer775-PFKL antibody or 1 µg of the isotype control overnight at 4 °C. Protein A beads (Pierce Protein A Agarose, Thermo Fisher Scientific) were prewashed 3 times with the lysis buffer. 100 µl of protein A resin slurry per sample was added to the antigen-antibody complex. After 2 h of incubation at room temperature, supernatant was collected as flow-through samples. Beads were washed 5 times with PBS, resuspended in 50 µl of 2 x Laemmli buffer, boiled at 95 °C for 5 min, and centrifuged. The supernatant was taken as immunoprecipitated samples.

## Enzyme activity assay

Enzymatic reactions were performed in reaction buffer containing 50 mM Tris-HCl (pH 7.4), 100 mM KCl, 10 mM MgCl$_2$, 1 mM DTT, 2 mM F6P, 2 mM ATP and 50 ng/µl PFKL$^{WT}$ or PFKL$^{S775A}$ protein at 37 °C for 2, 4, 6, 8, and 10 min and stopped by the addition of 20 µl of trifluoroacetic acid (pH 2.0). F1,6BP production was quantified by LC-MS. Because the PFKL kinase reaction rate was linear for the first 2 min, the amount of F1,6BP formed per min during the first 2 min was used to determine the initial reaction rate.

## LC-MS analysis of F1,6BP

Samples were diluted in mobile phase consisting of 10% acetonitrile and 90% 20 mM ammonium formate buffer (pH 3.0). 15 µl of the samples were transferred to an Ultimate 3000 chromatography system (Thermo Fisher Scientific) and separated on a Newcrom B HPLC column (0.5 × 100 mm, 5 µm, 100 A, SIELC Technologies Wheeling, IL USA). Separation was performed at an isocratic flow rate of 75 µl/min for 15 min. A TripleTOF 5600 mass spectrometer (SCIEX, Toronto, Canada) was used for detection. The instrument was operated in negative ion mode with the following scan parameters: Scan type TOF-MS; mass range 200–400 Da; accumulation time 1 s; declustering potential -50. Chromatographic peaks were quantified using PeakView V2.2 software (SCIEX) at 259.0 m/z for F6P and 338.9 m/z for F1,6BP.

## In vitro PKC δ kinase assay

The PKCδ kinase assay was performed following the manufacturer's instructions (Promega). Briefly, serially diluted PKCδ was incubated with 2 µM purified PFKL$^{WT}$ or PFKL$^{S775A}$ in the present of lipid activator, 50 µM ATP and 50 µM DTT at room temperature for 60 minutes. The reaction was stopped by the addition of 2 x Laemmli buffer.

## Flow Cytometry

Bone marrow cells derived from 10-12-week-old, male C57BL/6 J WT and Pfkl$^{S775A/S775A}$ mice were flushed from the femur and tibia into Ca$^{2+}$- and Mg$^{2+}$-free PBS using a syringe. Clumps and debris in the cell suspensions were removed by passing the cell suspension through a 100 µm cell strainer. Cells were spun down at 300 g for 10 min and the cell pellet was resuspended in 2 ml 1 x RBC lysis buffer (BioLegend) for 2 min at room temperature to get rid of red blood cells. After centrifugation, cells were resuspended in FACS buffer (PBS with 2% FCS). Single splenocyte suspension was obtained by thoroughly mincing the spleen with a scissor, followed by passage through a 100 µm cell strainer. Red blood cells were depleted by incubating splenocytes with 1 x RBC lysis buffer for 2 min at room temperature. Cells were then washed with FACS buffer and blocked with mouse FcR blocking reagent (Miltenyi Biotec) for 20 min at 4 °C to avoid unspecific antibody binding. For cell surface staining, cells were stained with 100 µl of

appropriate antibodies against surface antigens on ice for 30 min in the dark. The antibodies used for FACS included APC/Cyanine7 anti-mouse CD3 (145-2C11; 1:50; BioLegend 100330), PE/Dazzle 594 anti-mouse CD4 (GK 1.5; 1:100; BioLegend 100456), PerCP/Cyanine5.5 anti-mouse CD8b.2 (53-5.8; 1:100; BioLegend 140418), PE/Cyanine7 anti-mouse CD19 (6D5; 1:200; BioLegend 115520), Brilliant Violet 421 anti-mouse CD335 (NKp46) (29A1.4; 1:50; BioLegend 137612), Alexa Fluor 488 anti-mouse/human CD11b (M1/70; 1:200; BioLegend 101217), Brilliant Violet 785 anti-mouse F4/80 (BM8; 1:500; BioLegend 123141), Alexa Fluor 700 anti-mouse CD11c (N418; 1:200; BioLegend 117320), APC anti-mouse I-A$^b$ (AF6-120.1; 1:50; BioLegend 116418), and Brilliant Violet 605 anti-mouse Ly-6G (1A8; 1:200; BioLegend 127639). Aqua fluorescent reactive dye (Invitrogen) was used according to the introduction from the manufacturer to exclude dead cells. Fluorescence minus one (FMO) controls were used to distinguish positively from negatively staining cell populations. Compensation was performed with Compbeads (BD). Flow cytometry was performed on a BD LSR Fortessa flow cytometer and data were acquired using BD FACS-Diva software. Data were analyzed using FlowJo software. Exemplary gating strategies are shown in Supplementary Figs. 7 and 8.

## TNF injection model

Recombinant mouse TNF (R&D Systems) was injected intraperitoneally to 10–12-week-old, male C57BL/6 J WT and Pfkl$^{S775A/S775A}$ mice to induce peritonitis, as described earlier[47]. 4 h later, mice were anaesthetized through isoflurane exposure and blood collected via retro orbital bleeding to assess white blood cell counts using the IDEXX flow cytometer (Westbrook, Maine, USA). In parallel, serum was collected from whole blood and cytokine measurements performed as follows. All mice were killed by cervical dislocation.

## Measurement of serum cytokines

Cytokines were measured using LEGENDplex™ mouse inflammation or antiviral panel with V-bottom plate (BioLegend) according to the manufacturer's instructions. Briefly, diluted serum or standards were incubated with capture beads coated with antibodies for 2 h at room temperature. The beads were washed once and incubated with detection antibodies for another hour at room temperature. SA-PE was then directly added to each well. After 30 min of incubation at room temperature, the beads were washed once, resuspended in wash buffer, and analyzed by flow cytometry.

## Detection of respiratory burst by Seahorse assay

mBMDM ($7 \times 10^4$ cells per well) were seeded in Seahorse 96-well cell culture plates and primed with 20 ng/ml mouse IFNγ (ImmunoTools) 1 day before the experiment. Next day, mBMDM were washed 3 times with Seahorse Assay DMEM medium supplemented with 10 mM glucose, 2 mM glutamine, and 2 mM pyruvate (all from Agilent Technologies), and then incubated at 37 °C for 45 min in a CO$_2$-free incubator. Real-time respiratory rates were measured in response to sequential injections by a Seahorse XFe96 Analyzer.

## Bacterial killing assay

mBMDM were plated in antibiotic-free DMEM medium containing 20 ng/ml mouse IFNγ in 24-well plates at a density of $5 \times 10^5$ per well. Next day, E.coli was subcultured to reach OD 0.5–0.6 and added to the cells at a MOI of 10. mBMDM were then incubated for 30 min at 37 °C to allow internalization. After 30 min, cells were washed 3 times with PBS, and DMEM medium containing 200 µg/ml gentamicin (Thermo Fisher Scientific) was added to kill extracellular bacteria. 1 h later, the medium was replaced with fresh DMEM containing 50 µg/ml gentamicin to block the division of extracellular bacteria. At the indicated time points (time after internalization), cells were washed once with PBS and lysed with 500 µl 0.05% Triton-x 100 (Sigma-Aldrich) in sterile water for 10 min at room temperature. Lysates were serially diluted in

PBS and plated on antibiotic-free agar plates and colonies were counted after overnight incubation at 37 °C.

### NADPH/NADP⁺ detection assay

The ratio of NADPH/ NADP⁺ was calculated after measuring the individual concentrations of NADPH and NADP⁺ with NADP/NADPH-Glo (Promega). mBMDM ($2 \times 10^4$ cells per well) were seeded in 96-well cell culture plates and primed with 20 ng/ml mouse IFNγ one day before the experiment. The next day cells were pretreated with DMSO or 10 μM DPI for 30 min, followed by zymosan treatment for 30 or 60 min. Cells were lysed in plates with 1% dodecyltrimethylammonium bromide. NADPH and NADP⁺ were measured according to the manufacturer's instructions.

### LC-MS analysis of cellular metabolites

mBMDM ($2 \times 10^6$ cells per well) were plated in 6-well plates and primed overnight with 20 ng/ml mouse IFNγ in normal DMEM culture medium and stimulated the next day with 100 μg/ml zymosan (Invivogen) in glucose-free DMEM medium supplemented with 10% dialyzed FCS (Thermo Fisher Scientific), 10 mM U-¹³C-glucose (Biomol), 100 U/ml Penicillin-Streptomycin, and 1 mM sodium pyruvate. After stimulation, the medium was removed and 1 ml of ice-cold 80% methanol: water solution (v/v), containing Maltose (Sigma-Aldrich) as an internal standard, was added to each well. After 20 min incubation at −80 °C, cells were scraped and cell suspension was centrifuged at $13,000 \times g$ for 10 min at 4 °C, and then supernatants were dried in a SpeedVac concentrator (Savant, Thermo). Lyophilized samples were reconstituted in 40 μl of LC/MS-grade water, vortexed, and centrifuged again for 10 min at $13,000 \times g$ at 4 °C. Samples were analyzed using an Agilent 1200 series HPLC system interfaced with an ABSciex 5500 hybrid triple quadrupole/linear ion trap mass spectrometer equipped with an electrospray ionization source operating in the positive or negative mode. The Q1 (precursor ion) and Q3 (fragment ion) transitions, the metabolite identifier, dwell times, and collision energies for both positive and negative ion modes were used according to published methods with additional transitions for our internal standards[48,49]. Five microliters of sample were injected into an XBridge Amide HPLC column (3.5 μm; $4.6 \times 100$ mm, 186004868, Waters, Milford, MA). The mobile phases were run at 400 μl/min and consisted of HPLC buffer A (pH = 9.0, 95% (v/v) water, 5% (v/v) acetonitrile, 20 mM ammonium hydroxide, 20 mM ammonium acetate) and HPLC buffer B (100% acetonitrile). The HPLC settings were as follows: from 0 to 0.1 min, the mobile phase was maintained at 85% buffer B; from 0.1 to 3 min, the percentage of buffer B was decreased from 85% to 30%; from 3 to 12 min, the percentage of buffer B was decreased from 30% to 2% and was maintained at 2% for an additional 3 min. At minute 15, the percentage of buffer B was increased again to 85%, and the column was flushed for an additional 8 min with 85% buffer B. Analyst (ABSciex) software was used for data analysis. Metabolite peaks were normalized by cell number and internal standards before statistical analyses. The retention time for all metabolites was verified using individual purified standards from Sigma: (Glycolysis/Gluconeogenesis Metabolite Library (ML0013-1KT), Pentose Phosphate Metabolite Library (ML0012), TCA Cycle Metabolite Library (ML0010)), L-glutathione reduced (G4251), L-glutathione oxidized (G6654), L-serine (S4500), and α-D-glucose 1-phosphate (G6750) using the same chromatographic method. The metabolites were quantified by integrating the chromatographic peak area of the precursor ion.

### qPCR

RNA from WT and *Pfkl*^S775A/S775A mBMDM was isolated and purified using the Total RNA Purification Mini Spin kit (Genaxxon) according to the supplier's protocols, followed by DNAse I treatment (Thermo Fisher Scientific). cDNA synthesis was performed using RevertAid reverse transcriptase (Thermo Fisher Scientific) with oligo dT primers. qPCR reaction was carried out using Takyon No ROX SYBR 2X MasterMix blue dTTP (Eurogentec) in the presence of primers specific for genes of interest.

### ELISA

mIL-6 and mTNF ELISA kits (BD) were used according to the supplier's protocols.

### Gene expression data

Expression of *Pfkl*, *Pfkm* and *Pfkp* as presented in Fig. 1d was determined by analyzing the publicly available dataset GSE124829 available on Immgen.org as normalized gene counts. Expression data of the 11 available immune cells subsets of male mice that were not further treated were summarized and are presented as mean values.

### Statistics & reproducibility

All statistical analyses were performed with GraphPad Prism 9. No statistical method was used to predetermine sample size. The exact number of replicates and statistical tests are indicated in the figure legends. Unless otherwise indicated, *n* represents the number of biological replicates using cells from different mice or independent cultures. A ROUT outlier correction was conducted for the in vivo cytokine measurements (Fig. 4i, j).

### Reporting summary

Further information on research design is available in the Nature Portfolio Reporting Summary linked to this article.

## Data availability

Publicly available gene expression data of murine immune cells were obtained from Immgen under the accession number GSE124829. Source data of the immunoblots are provided with this paper. Other information is available in the Supplementary Information and otherwise available upon request. Source data are provided with this paper.

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

## Acknowledgements

We kindly thank C. Ludwig from the BioSysM flow cytometry facility for great support. This work was supported by grants from the Deutsche Forschungsgemeinschaft (DFG, German Research Foundation) CRC 1403/A03 (Project-ID 414786233) to V.H.

## Author contributions

Conceptualization: M.W. and V.H.; Data curation: V.H.; Formal analysis: M.W., A.J., M.N., S.M.A., J.M.K., A.Henne, A.Heinz, M.B., T.F., and V.H.; Funding acquisition: V.H.; Investigation: M.W., H.F., A.J., M.N., S.M.A., J.M.K., A.Henne, A.Heinz, M.B., E.K., and T.F.; Methodology: M.W., H.F., A.J., M.N., S.M.A., J.M.K., A.Henne, A.Heinz, M.B., N.A.S., E.K., B.W., W.W.,

M.S., J.R., and T.F.; Project administration: V.H.; Resources: K.H., B.W., W.W., M.S., J.R., and T.F.; Supervision: K.H.; Visualization: M.W. and V.H.; Writing – original draft: M.W. and V.H.; Writing - review & editing: M.W., H.F., A.J., M.N., S.M.A., J.M.K., A.Henne, A.Heinz, M.B., N.A.S., K.H., E.K., B.W., W.W., M.S., J.R., T.F., and V.H.

## Funding

## Competing interests
The authors declare no competing interests.
