## [Peer Review File · Nature Communications]

Phosphorylation of PFKL regulates metabolic reprogramming in macrophages following pattern recognition receptor activationEditorial Note: This manuscript has been previously reviewed at another journal. This document only contains reviewer comments and rebuttal letters for versions considered at *Nature Communications*. Mentions of prior referee reports have been redacted.

REVIEWERS' COMMENTS

Reviewer #1 (Remarks to the Author):

The authors have adequately addressed my concerns and I'm happy to recommend acceptance.

Reviewer #2 (Remarks to the Author):

We reviewed this study previously for [REDACTED]. In fact, the authors provide a modified version to address our initial concerns. However, the strength of the study is that they have the phospho-mutant mouse strain, and that they also have human monocyte-derived macrophage data supporting their findings. But since glycolysis-dependency in macrophages is so well-known, and the authors are merely adding the Pfkfb3 phosphorylation as an important phosphorylation in the glycolytic pathway in macrophages, we have concerns on the novelty of this study. Here are the two specific questions we have regarding to this updated manuscript.

1. They used their Pfkfb3 mutant mice strain to show that PFKFB3 phosphorylation is required for macrophage responses to different PRR agonists. They originally claimed that the mutant increased its flux into PPP, resulting in lower ROS and less bactericidal activity, but we asked additional experiments to prove this part, that they did and now they removed the PPP and ROS part because it didn't support their claim.
2. The original manuscript was based predominantly on BMDM data, so we requested they used their mouse model for in vivo proof that it had physiological effect. They now did a TNF-induced inflammation model. By injecting TNF i.p. they see lower serum-levels of MCP1 and IL1b, but the data are not super striking although they are significant (Figure 4i+j),

REVIEWERS' COMMENTS

Reviewer #1 (Remarks to the Author):

The authors have adequately addressed my concerns and I'm happy to recommend acceptance.

We appreciate the reviewer's thorough evaluation of our manuscript and are delighted to hear that they consider it suitable for publication.

Reviewer #2 (Remarks to the Author):

We reviewed this study previously for [REDACTED]. In fact, the authors provide a modified version to address our initial concerns. However, the strength of the study is that they have the phospho-mutant mouse strain, and that they also have human monocyte-derived macrophage data supporting their findings. But since glycolysis-dependency in macrophages is so well-known, and the authors are merely adding the Pfk1 phosphorylation as an important phosphorylation in the glycolytic pathway in macrophages, we have concerns on the novelty of this study. Here are the two specific questions we have regarding to this updated manuscript.

1. They used their Pfk1S775A phospho-mutant mice strain to show that PFKL phosphorylation is required for macrophage responses to different PRR agonists. They originally claimed that the mutant increased its flux into PPP, resulting in lower ROS and less bactericidal activity, but we asked additional experiments to prove this part, that they did and now they removed the PPP and ROS part because it didn't support their claim.

In the initial version of our manuscript, we demonstrated that *Pfk1*^{S775A/S775A} macrophages generated more ROS compared to WT macrophages. Given that PFKL Ser775 phosphorylation enhances its catalytic activity and PFKL activity negatively regulates ROS production through regulating NADPH production as indicated by existing literature, we hypothesized that more glucose might be shuttled through the pentose phosphate pathway (PPP) in *Pfk1*^{S775A/S775A} macrophages, thereby contributing to NADPH production and subsequent ROS production. Following the reviewer's suggestions, we conducted additional experiments to test this hypothesis. Our findings revealed only slight difference in NADPH levels between WT and *Pfk1*^{S775A/S775A} macrophages, suggesting that additional mechanisms, beyond NADPH production, may contribute to the observed differences in ROS levels between WT and *Pfk1*^{S775A/S775A} macrophages. The data related to ROS and NADPH are now included in Supplementary Fig. 5 and presented in the results section of the manuscript.

2. The original manuscript was based predominantly on BMDM data, so we requested they used their mouse model for in vivo proof that it had physiological effect. They now did a TNF-induced inflammation model. By injecting TNF i.p. they see lower serum-levels of MCP1 and IL1b, but the data are not super striking although they are significant (Figure 4i+j).

In order to minimize confounding factors and achieve a clearer understanding of the genuine impact of the glycolysis on macrophage function and immune responses, ultimately providing more reliable insights into the underlying biology, we developed a genetic model to examine a specific metabolic enzyme's role (PFKL Ser775 phosphorylation) in macrophage function without disrupting it. It is important to note that focusing on a single post-translational modification in a complex metabolic pathway carries the risk of redundancy, as multiple regulatory steps might affect glycolysis. With these considerations in mind, we suggest that the moderate but significant quantitative differences observed in relation to PFKL phosphorylation are likely attributable to its role in glycolysis.